# Alternative splicing of *coq-2* controls the levels of rhodoquinone in animals

June H Tan[1], Margot Lautens[1], Laura Romanelli-Cedrez[2], Jianbin Wang[3,4], Michael R Schertzberg[1], Samantha R Reinl[5], Richard E Davis[3], Jennifer N Shepherd[5]*, Andrew G Fraser[1]*, Gustavo Salinas[2]*

[1]The Donnelly Centre, University of Toronto, Toronto, Canada; [2]Laboratorio de Biología de Gusanos. Unidad Mixta, Departamento de Biociencias, Facultad de Química, Universidad de la República - Institut Pasteur de Montevideo, Montevideo, Uruguay; [3]Department of Biochemistry and Molecular Genetics, RNA Bioscience Initiative, University of Colorado School of Medicine, Aurora, United States; [4]Department of Biochemistry and Cellular and Molecular Biology, University of Tennessee, Knoxville, United States; [5]Department of Chemistry and Biochemistry, Gonzaga University, Spokane, United States

*For correspondence:
shepherd@gonzaga.edu (JNS);
andyfraser.utoronto@gmail.com
(AGF);
gsalin@fq.edu.uy (GS)

Competing interests: The authors declare that no competing interests exist.

**Abstract** Parasitic helminths use two benzoquinones as electron carriers in the electron transport chain. In normoxia, they use ubiquinone (UQ), but in anaerobic conditions inside the host, they require rhodoquinone (RQ) and greatly increase RQ levels. We previously showed the switch from UQ to RQ synthesis is driven by a change of substrates by the polyprenyltransferase COQ-2 (Del Borrello et al., 2019; Roberts Buceta et al., 2019); however, the mechanism of substrate selection is not known. Here, we show helminths synthesize two *coq-2* splice forms, *coq-2a* and *coq-2e*, and the *coq-2e*-specific exon is only found in species that synthesize RQ. We show that in *Caenorhabditis elegans* COQ-2e is required for efficient RQ synthesis and survival in cyanide. Importantly, parasites switch from COQ-2a to COQ-2e as they transit into anaerobic environments. We conclude helminths switch from UQ to RQ synthesis principally via changes in the alternative splicing of *coq-2.*

## Introduction

Parasitic helminths are major human pathogens. Soil-transmitted helminths (STHs) such as *Ascaris*, hookworms, and whipworms infect well over a billion people, and 7 out of the 18 neglected diseases categorized by WHO are caused by helminths (*CDC, 2019*). Despite the huge impact on global health of these infections, there are few classes of available anthelmintics and resistance is increasing in humans and is widespread in some species that infect animals — for example, *Haemonchus contortus*, a common parasite in small ruminants, has developed multidrug resistance (*Jackson and Coop, 2000*; *Kotze and Prichard, 2016*). Thus, there is a serious need to develop new classes of anthelmintics that target the parasites while leaving their animal hosts unaffected. One potential target for anthelmintics is their unusual anaerobic metabolism which differs from that of their hosts. When STHs are in the free-living stages of their life cycles, they use the same aerobic respiration as their hosts and use ubiquinone (referred to here as UQ and as Q in other papers) as an electron carrier in their mitochondrial electron transport chain (ETC). However, when they infect their hosts, they encounter highly anaerobic environments. This is particularly the case for species that inhabit the host gut, for example *Ascaris* can live for many months in this anaerobic environment (*Dold and Holland, 2011*). To survive, they use an alternate form of anaerobic metabolism that relies on the electron carrier rhodoquinone (RQ). Since hosts do not synthesize or use RQ, RQ-dependent

metabolism could be a key pharmacological target as it is required by the parasite but absent in mammalian hosts.

RQ is an electron carrier that functions in the mitochondrial ETC of STHs (*Van Hellemond et al., 1995*). RQ is a prenylated aminobenzoquinone that is similar to UQ (*Figure 1*), but the slight difference in structure gives RQ a lower standard redox potential than UQ ($-63$ mV and $110$ mV, respectively) (*Erabi et al., 1976*; *Unden and Bongaerts, 1997*). This difference in redox potential means that RQ, but not UQ, can play a unique role in anaerobic metabolism. In aerobic metabolism, UQ can accept electrons from a diverse set of molecules via quinone-coupled dehydrogenases, including succinate dehydrogenase and electron-transferring-flavoprotein (ETF) dehydrogenase. In RQ-dependent anaerobic metabolism, RQ does the reverse — it carries electrons

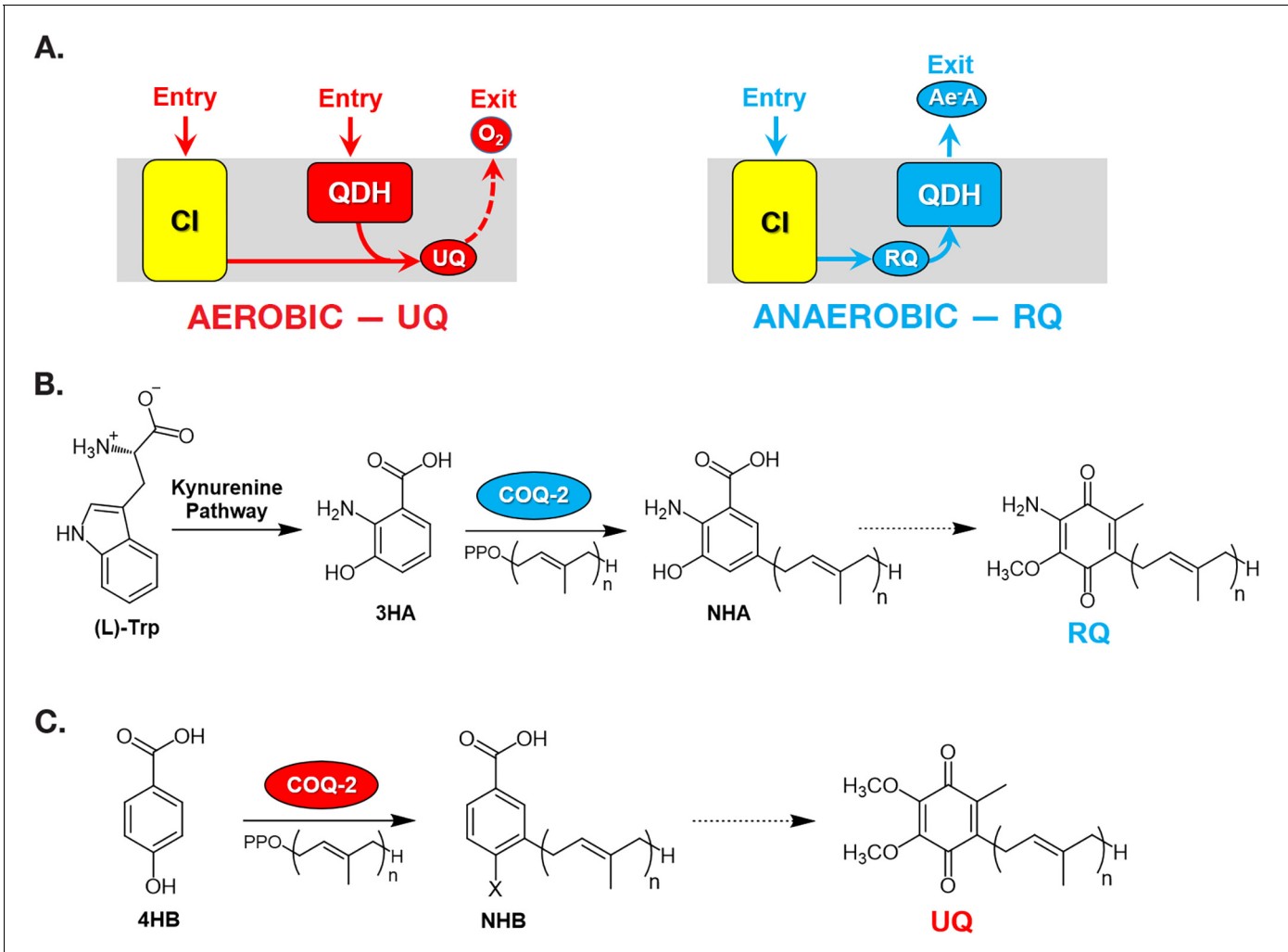

**Figure 1.** Rhodoquinone and ubiquinone biosynthesis and function in electron transport chains. (A) In aerobic metabolism, ubiquinone (UQ) shuttles electrons in the ETC from Complex I (CI; yellow box) and quinone-coupled dehydrogenases (QDHs), such as Complex II. These electrons are ultimately transferred to oxygen. In anaerobic metabolism, rhodoquinone (RQ) reverses electron flow in QDHs and facilitates an early exit of electrons from the ETC onto anaerobic electron acceptors (Ae⁻A), such as fumarate. (B) The RQ biosynthetic pathway in *C. elegans* requires *L*-tryptophan, a precursor in the kynurenine pathway. *L*-Trptophan is transformed into 3-hydroxyanthranilic acid (3HA) in four steps. It is proposed that 3HA is a substrate for COQ-2, producing 3-hydroxy-5-nonaprenylanthranilic acid (NHA), where n=9. The transformation of NHA to RQ requires several shared proteins from the UQ biosynthetic pathway. (C) Eukaryotes can use either *p*-aminobenzoic acid (pABA) or 4-hydroxybenzoic acid (4HB) as precursors to UQ. Prenylation is facilitated by Coq2 to form 3-hexaprenyl-4-hydroxybenzoic acid (HHB) or 3-hexaprenyl-4-aminobenzoic acid (HAB), where n = varies between species. Further functionalization of these intermediates occurs through a Coq synthome (*Coq*3–Coq9 and Coq11) to yield UQ.

to these same dehydrogenase enzymes and drives them in reverse to act as reductases that transfer electrons onto a diverse set of terminal acceptors (*Tielens and Van Hellemond, 1998*; *van Hellemond et al., 2003*). UQ thus allows electrons to enter the ETC via dehydrogenases; RQ can drive the reactions that let electrons leave the ETC and this difference in the direction of electron flow is driven by the difference in redox potential (*Figure 1A*). This ability of RQ to provide electrons to an alternative set of terminal electron acceptors allows helminths to continue to use a form of mitochondrial ETC to generate ATP without oxygen. In this RQ-dependent anaerobic metabolism, electrons enter the ETC from NADH through Complex I and onto RQ, and this is coupled to proton pumping to generate the proton motive force required for ATP synthesis by F0F1-ATPase (*van Hellemond et al., 2003*). The electrons are carried by RQ to the quinone-coupled dehydrogenases which are driven in reverse as reductases and the electrons thus exit onto a diverse set of terminal acceptors, allowing NAD$^+$ to be regenerated and the redox balance to be maintained. This RQ-dependent metabolism does not occur in the hosts and is thus an excellent target for anthelmintics (*Kita et al., 2003*).

In the animal kingdom, RQ is present in several facultative anaerobic lineages that face environmental anoxia or hypoxia as part of their life cycle. Among animals, RQ has only been described in nematodes, platyhelminths, mollusks, and annelids (*Van Hellemond et al., 1995*). The key steps of RQ biosynthesis in animals were recently elucidated (*Roberts Buceta et al., 2019*; *Del Borrello et al., 2019*). In contrast to bacteria, where RQ derives from UQ (*Bernert et al., 2019*; *Brajcich et al., 2010*), RQ biosynthesis in animals requires precursors derived from tryptophan (*Figure 1B*). Recent studies performed on *C. elegans* have demonstrated that animals that lack a functional kynureninase pathway (e.g. strains carrying mutations in *kynu-1,* the sole kynureninase) are unable to synthesize RQ (*Del Borrello et al., 2019*; *Roberts Buceta et al., 2019*). It is presumed that 3-hydroxyanthranilic acid (3HA, also sometimes referred to as 3HAA) from the kynurenine pathway is prenylated in a reaction catalyzed by COQ-2. The prenylated benzoquinone ring can then be modified by methylases and hydroxylases to form RQ. This proposed pathway is analogous to the biosynthesis of UQ from 4-hydroxybenzoic acid (4HB) or *para*-aminobenzoic acid (pABA) (*Figure 1C*). The key insight from these previous studies is that the critical choice between UQ and RQ synthesis is the choice of substrate by COQ-2 — if 4HB is prenylated by COQ-2, UQ will ultimately be produced, but if 3HA is used, the final product will be RQ. In most parasitic helminths, there is a major shift in quinone composition as they move from aerobic to anaerobic environments, for example, RQ is less than 10% of *H. contortus* total quinone when the parasite is in an aerobic environment but is over 80% of total quinone in the anaerobic environment of the sheep gut and similar shifts occur in other parasites (*Lümmen et al., 2014*; *Sakai et al., 2012*; *Van Hellemond et al., 1995*). Somehow, COQ-2 must, therefore, switch from using 4HB to using 3HA as a substrate, but the mechanism for this substrate switch is completely unknown. Understanding this mechanism is important — if we could interfere pharmacologically with the switch to RQ synthesis, it could lead to a new class of anthelmintics.

In this study, we reveal that two variants of COQ-2, derived from alternative splicing of mutually exclusive exons, are the key for the choice to synthesize RQ or UQ. We find that one of the mutually exclusive exons is only found in the genomes of animals that synthesize RQ and that its inclusion remodels the core of the COQ-2 enzyme. We show that the removal of this exon abolishes almost all RQ biosynthesis in *C. elegans*. Finally, we find that inclusion of this RQ-specific exon expression is increased in the stages of the parasite life cycle where they encounter hypoxic conditions and increase RQ production, while the alternative exon is increased in normoxic life stages. We thus conclude that alternative splicing of COQ-2 is the key mechanism that causes the switch from UQ to RQ synthesis in the parasite life cycle.

## Results

### The *C. elegans* COQ-2 polyprenyl transferase required for quinone biosynthesis has two major alternative splice forms

Our research groups previously showed that if COQ-2 uses 4HB as a substrate, it would lead to the synthesis of UQ; however, if COQ-2 uses 3HA, it would ultimately yield RQ. As parasites move from aerobic environments to the anaerobic niches in their host, they change their quinone

composition from high UQ to high RQ. For this to occur, COQ-2 must switch its substrate from 4HB to 3HA, but the mechanism for this switch is unknown.

We identified two distinct splice forms of *C. elegans coq-2*: *coq-2a* and *coq-2e*. These are annotated in the genome and confirmed by RNA-sequencing by nanopore-based direct RNA sequencing, and by targeted validation studies (*Kuroyanagi et al., 2014*; *Ramani et al., 2011*; *Roach et al., 2020*). These two isoforms differ by the mutually exclusive splicing of two internal exons (6a and 6e), both of 134 nucleotides (see *Figure 2A*). We note that mutually exclusive splicing of cassette exons is very rare in the *C. elegans* genome and fewer than 100 such splicing events have been identified (*Kuroyanagi et al., 2014*; *Ramani et al., 2011*). Both *coq-2a* and *coq-2e* splice forms are abundant in *C. elegans* across all stages of development, where ~30–50% of *coq-2* is the *coq-2e* isoform (*Gerstein et al., 2010*; *Grün et al., 2014*; *Ramani et al., 2011*; *Roach et al., 2020*).

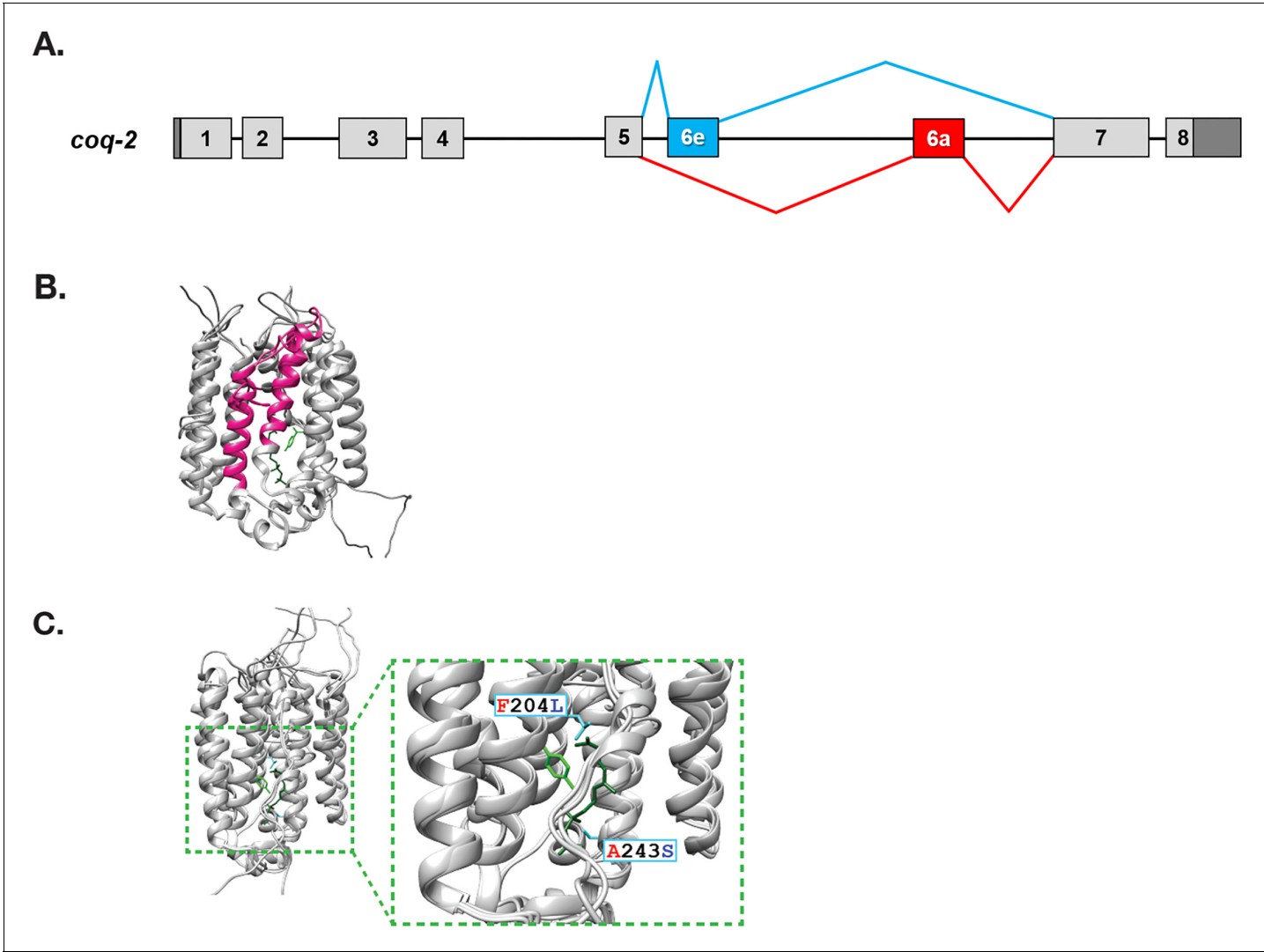

**Figure 2.** *C. elegans coq-2* gene model. (**A**) The *coq-2* gene contains two mutually exclusive exons, 6e (blue box) and 6a (red box), that are alternatively spliced (blue and red lines, respectively) generating two COQ-2 isoforms. Light gray boxes represent coding sequences of exons 1–5 and 7–8, black lines represent introns, and dark gray boxes denote 5′and 3′ untranslated regions of exons 1 and 8. (**B**) Alternative splicing of COQ-2 changes the enzyme core. The sequences of *C. elegans coq-2a* and *coq-2e* were threaded onto the crystal structure of the apo-form of the *Aeropyrum pernix* COQ-2 homolog (PDB: 4OD5) in Chimera using Modeller. The region switched by mutually exclusive alternative splicing is shown magenta color. (**C**) The alternative exons found in all RQ-synthesizing species have two residues that are invariant (L204 and S243 show in cyan; *C. elegans* numbering) that are near the binding site of the two substrates. Substrates are the polyprenyl tail (dark green; geranyl S-thiolodiphosphate in the crystal structure), and the aromatic ring (light green; *p*-hydroxybenzoic acid [4HB] in the crystal structure). Note that COQ-2 is rotated from panel B to panel C for clarity.

To examine how the alternative splicing of COQ-2 might affect its function, we threaded the predicted COQ-2a and COQ-2e protein sequences onto the solved crystal structure of the archaean *A. pernix* COQ-2 ortholog (PDB:4OD5). We found that the splicing change causes a switch in two α-helices at the core of the COQ-2 structure (*Figure 2B*). This is a region of the protein that is believed to form a hydrophobic tunnel along which aromatic substrates must pass to the active site for the key polyprenylation reaction (*Desbats et al., 2016*), suggesting that the change in splicing could affect *COQ-2* substrate selection and thus could explain a shift from UQ to RQ synthesis. We, therefore, examined whether similar COQ-2 alternative splicing is seen in parasitic helminths and how the different splice forms compare to COQ-2 sequences in parasite hosts which do not synthesize RQ.

### Parasitic helminths have a distinct splice form of *coq-2* that is not present in any of the parasitic hosts

*C. elegans* has two major isoforms of *coq-2* which resulted from mutually exclusive alternative splicing of two internal exons and affects the core of the enzyme. If this alternative splicing affects the choice of COQ-2 substrate and thus the switch from UQ to RQ synthesis, we reasoned that parasitic helminths that synthesize RQ should have a similar gene structure and that the hosts that do not synthesize RQ should not. We used both gene predictions and available RNA-seq data to examine the

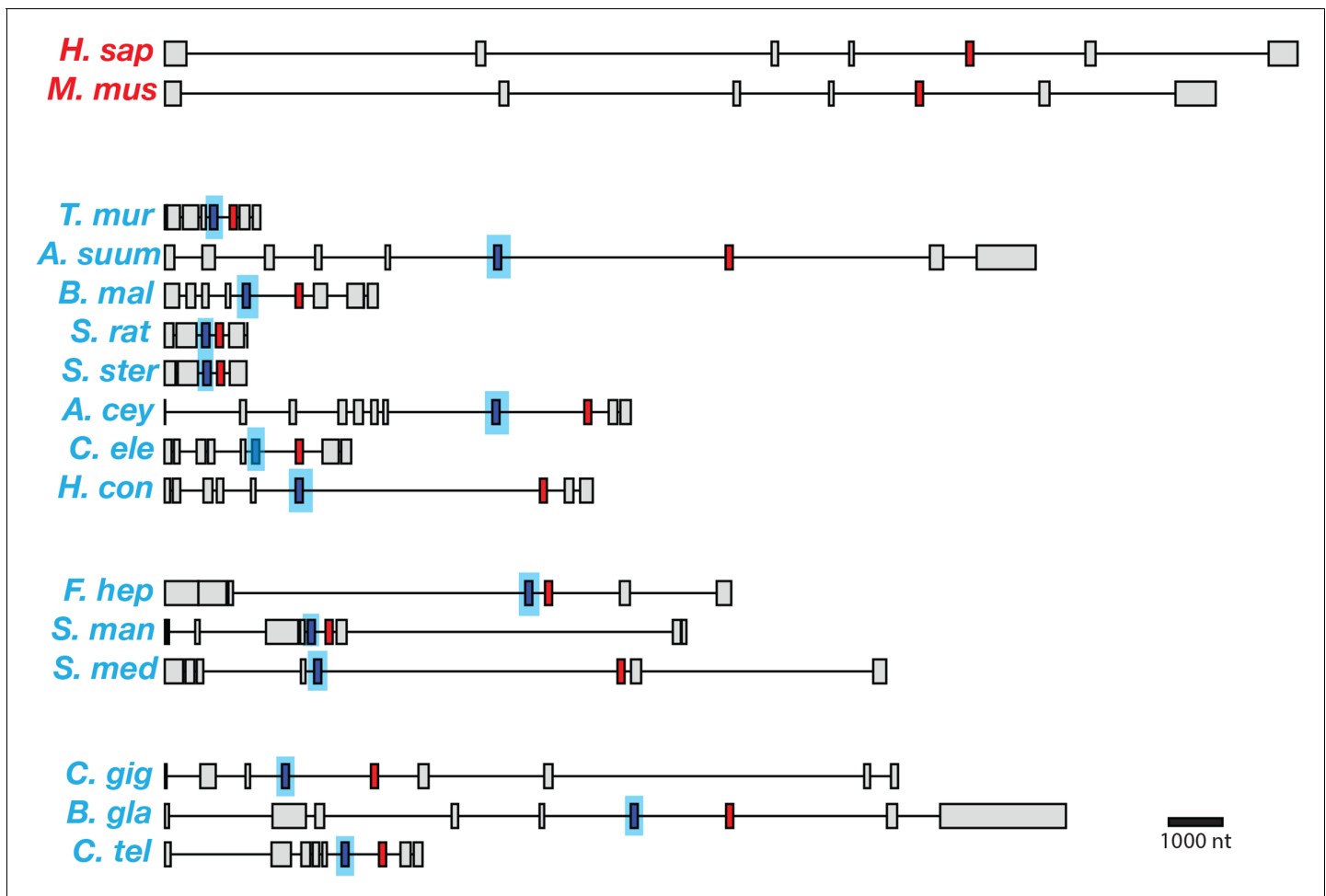

**Figure 3.** Gene models for *coq-2* orthologs in various species. Parasitic helminths, as well as annelids and mollusks, have two internal exons that are spliced in a mutually exclusive manner. By contrast, humans and other hosts only have one exon that is homologous to exon 6a of *C. elegans coq-2*. The a-form exon (red) shares greater similarity to the exon present in species that do not synthesize RQ, while the e-form (blue) is present only in RQ-producing species. The genes used for each species are listed in *Supplementary file 3*. The gene structures shown are based on genome annotations but in many cases include manual reannotations — in all such cases, the manual annotations are confirmed with RNA-seq data.

*coq-2* gene structure and splicing in parasitic helminths and their hosts (human, sheep, cow, and rat, respectively; *Figure 3*). Since many of the parasite gene structures were not correctly annotated, all the relevant exon junctions in *Figure 3* were manually annotated and confirmed with RNA-seq data (see Materials and methods). We note that while the alternative splicing of *coq-2* in *C. elegans* we observe using RNA-seq has been validated by nanopore-based direct transcript sequencing, micro-array analysis, and RT-PCR, the data on the other helminths only derives from RNA-seq and genome analysis. Although the same robust RNA-seq analysis pipeline has been used, it would be ideal to validate these changes in the future. Remarkably, we find a similar gene structure with the same mutually exclusive exons in all parasites known to synthesize RQ (*Figure 3*). Furthermore, annelids and mollusks are the only other phyla known to synthesize RQ and their *coq-2* orthologs also show similar mutually exclusive alternative splicing of homologous exons and we only find evidence for alternative splicing between a *coq-2a* form and a *coq-2e* like form in animals that synthesize RQ. Crucially, no mammalian hosts show any evidence for this kind of alternative splicing of their *coq-2* orthologs, nor do they have an e-like exon either in their gene predictions or genome sequences or in any available RNA-seq or direct transcriptome sequence data (human and mouse are shown as representatives in *Figure 3*). Note that the RNA-seq datasets are much deeper and more extensive in humans and mouse than in helminths, mollusks, or annelids, so this lack of evidence for such alternative splicing is unlikely to be due to a failure to detect such events due to low coverage transcriptome data. We conclude that the e-exon and the mutually exclusive alternative splicing of *coq-2* is unique to animals that synthesize RQ and is not found in any other species. This suggests that this *coq-2* alternative splicing could indeed be linked to the ability to synthesize RQ.

We aligned the two mutually exclusive exons across helminth species and compared them to the similar regions of their host COQ-2 sequences, and of other eukaryotes that cannot synthesize RQ (*S. cerevisiae* and *S. pombe*) (*Figure 4*). We find that all the *coq-2a*-specific exons are similar to the pan-eukaryotic COQ-2 sequence, whereas all the *coq-2e*-specific exons have a distinct sequence to this. We examined the alignments of the a- and e-specific exons and identified two residues that are strictly conserved in pan-eukaryotic COQ-2 sequences, Phe204 and Ala243 (*C. elegans* numbering), that are switched to a Leu and a Ser residue in all COQ2-e-specific exons that we examined (*Figure 4*). These residues sit very close to the substrates in the active site of the enzyme (*Figure 2C*) and we note that mutation of the equivalent Ala243 residue dramatically affects the ability of human COQ-2 to synthesize UQ (*Desbats et al., 2016*). Altogether these results suggest that animals that synthesize both UQ and RQ synthesize two forms of COQ-2 — one looks similar to that in all other eukaryotic species, whereas the other has a single exon that appears to be specific for species that synthesize RQ. To test whether these two COQ-2 isoforms have distinct roles in UQ and RQ synthesis, we turned to *C. elegans.*

## Efficient RQ synthesis requires the *coq-2e* isoform

We found that helminths synthesize two major isoforms of *coq-2*, whereas their hosts only synthesize a single isoform. To test the requirement for each of the two major helminth isoforms of *coq-2* for RQ synthesis, we used CRISPR engineering to generate *C. elegans* mutant strains that either lack *coq-2* exon 6a (*coq-2(syb1715)*) or *coq-2* exon 6e (*coq-2(syb1721)*) — we refer to these as *coq-2Δ6a* and *coq-2Δ6e*, respectively, from here on (see *Figure 5A* and *Supplementary file 1* for details of engineering). We find that the *coq-2Δ6e* strain synthesizes essentially no detectable RQ but has higher levels of UQ, whereas *coq-2Δ6a* has greatly reduced UQ levels but higher RQ levels (*Figure 5B*, *Supplementary file 2*). We conclude that the *coq-2e* isoform, which includes the helminth/annelid/mollusc-specific exon 6e, is required for efficient RQ synthesis. While these engineered changes could potentially affect RQ levels via some indirect mechanisms such as alterations in the levels of 4HB and 3HA, it is unclear what mechanism would drive those metabolic rewirings. We thus suggest that the more parsimonious explanation is the one we propose here that COQ-2a and COQ-2e have different substrate preferences due to the large change to the core of the enzyme.

To further examine whether COQ-2e is required for RQ synthesis and thus for RQ-dependent metabolism, we tested whether the *coq-2Δ6a* and *coq-2Δ6e* strains could survive long-term exposure to potassium cyanide (KCN) (*Figure 5C* and *Figure 5—figure supplement 1*). We previously showed that when *C. elegans* is exposed to KCN, it switches to RQ-dependent metabolism and that while wild-type worms can survive a 15 hr exposure to KCN, *C. elegans* strains that do not synthesize

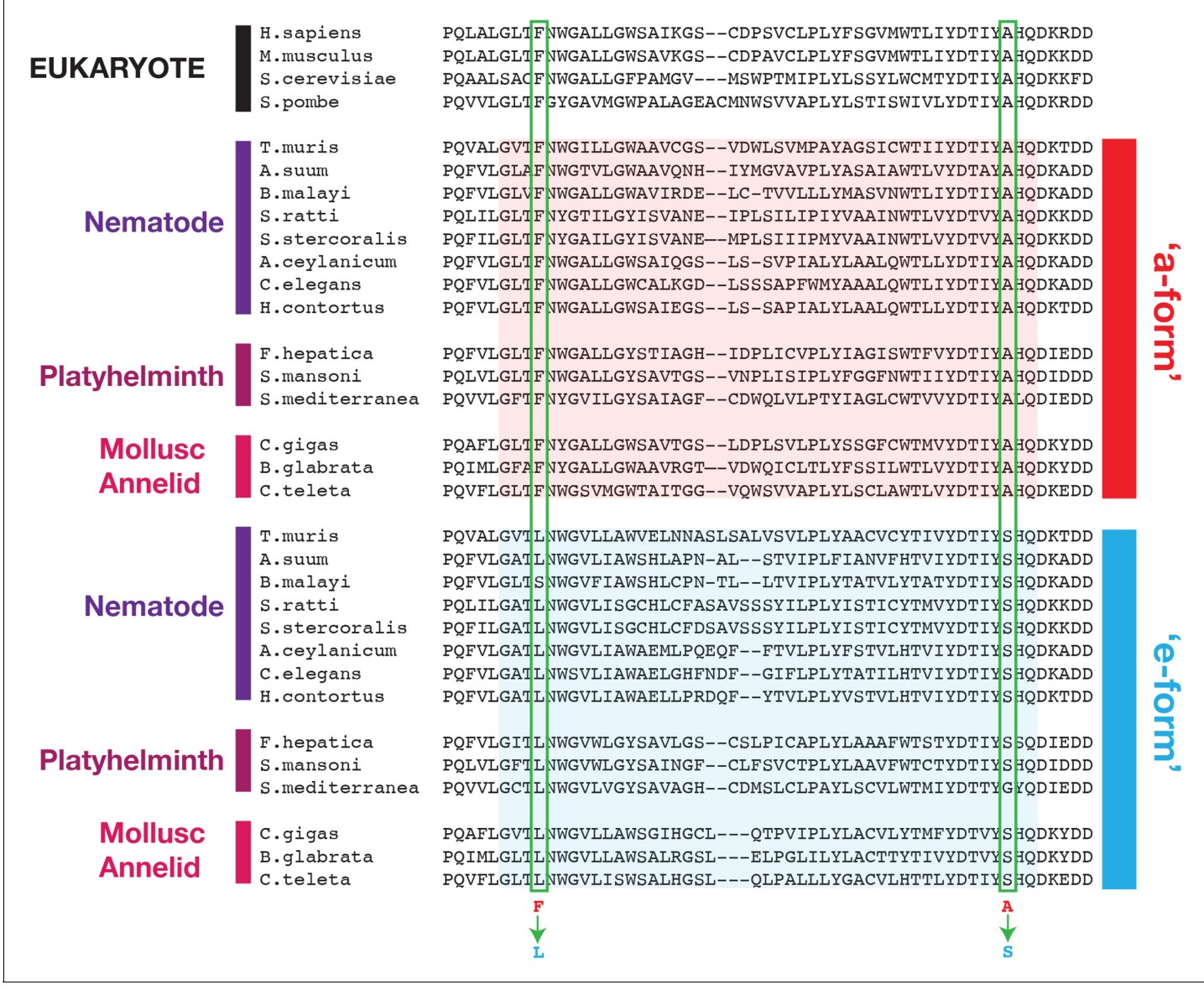

**Figure 4.** Conserved changes between a- and e-form exons across RQ-producing species. Amino acid sequences of COQ-2 orthologs were aligned using Clustal Omega (*Madeira et al., 2019*). The sequences of exons homologous to exon 6a/e in *C. elegans*, as well as the flanking five amino acid sequences, were used to generate the alignment. Sequences of the mutually exclusive exons are shaded in red (a-form) or blue (e-form). Two residue changes between the a- and e- forms are highlighted (Phe to Leu, Ala to Ser) and are invariant across diverse species that synthesize RQ. The COQ-2 orthologs and exons used for each species are listed in *Supplementary file 3*.

RQ cannot survive. We found that while the *coq-2Δ6a* strain (that can synthesize RQ) survives 15 hr of KCN exposure, as well as wild-type animals, the *coq-2Δ6e* strain that synthesizes no RQ does not survive, confirming the functional relevance of the *coq-2e* isoform as being critical for RQ synthesis.

## Regulation of the alternative splicing of *coq-2* in helminths

Helminths synthesize two isoforms of COQ-2 — COQ-2a resembles the pan-eukaryotic consensus and cannot synthesize RQ, whereas COQ-2e includes an exon that is only found in species that synthesize RQ and COQ-2e is required for RQ synthesis. Changing the levels of *coq-2a* and *coq-2e* splice forms could thus regulate the switch from UQ synthesis in the aerobic environment outside the host to RQ synthesis in the host gut. We thus examined RNA-seq data to see whether parasites switch between these isoforms as they switch between these environments.

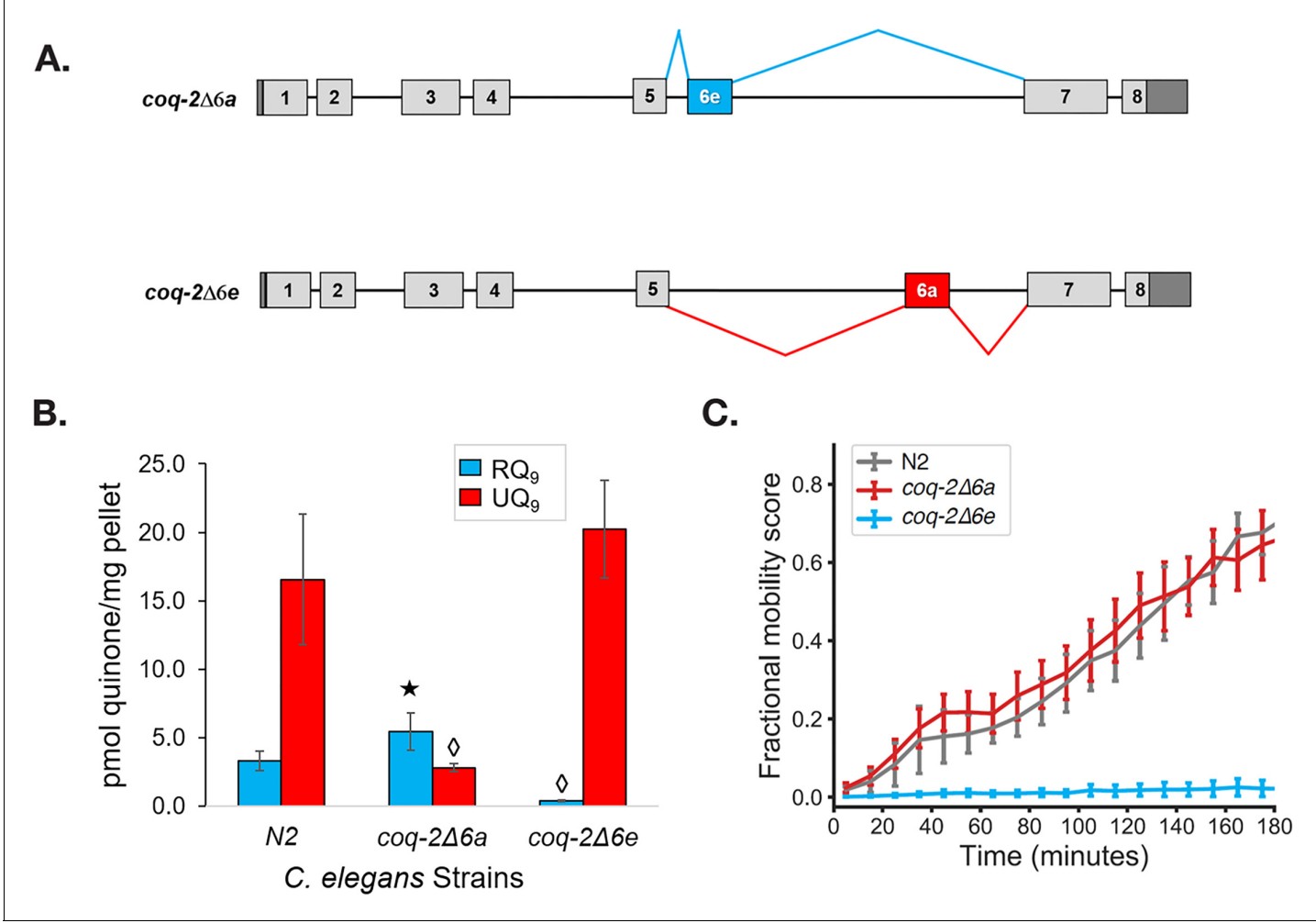

**Figure 5.** The *C. elegans coq-2* edited strains and effects of exon 6a and 6e deletions on quinone biosynthesis. (**A**) Mutant strains were generated in *C. elegans* by deletion of exon 6a (*coq-2Δ6a*) or exon 6e (*coq-2Δ6e*). (**B**) Deletion of exon 6a from the *coq-2* gene significantly increased the level of $RQ_9$ (p=0.013) and significantly decreased $UQ_9$ (p<0.001) compared to the N2 control. By contrast, the deletion of exon 6e decreased $RQ_9$ to a negligible level (p<0.001) and slightly increased the level of $UQ_9$ (p=0.130) compared to N2. Statistically significant increases and decreases with respect to N2 levels are denoted with ★ and ◊, respectively; error bars reflect standard deviation where N = 4. (**C**) Deletion of *coq-2* exon 6e affects the ability of worms to survive extended KCN treatment. Wild-type (**N2**) and *coq-2* mutant L1 worms were exposed to 200 μM KCN for 15 hr. KCN was then diluted 6-fold and worm movement was measured over 3 hr to track recovery from KCN exposure (see Materials and methods). Worms without exon 6e could not survive extended treatment with KCN while deletion of exon 6a had little effect on KCN survival. Cyanide titration is shown in *Figure 5—figure supplement 1*. Curves show the mean of four biological replicates and error bars are standard errors of the mean.

The online version of this article includes the following figure supplement(s) for figure 5:

**Figure supplement 1.** Deletion of *coq-2* exon 6e affects the ability of worms to survive extended KCN treatment at various KCN concentrations.

*Ascaris suum* has a relatively simple life cycle (schematic *Figure 6A*) and is the pig equivalent of *Ascaris lumbricoides* which infects ~900 M humans. Eggs are laid in the host and emerge via defecation and the L1-L3 larval stages develop within the egg outside the host. The L3 infective larval stage then enters the host via ingestion into the digestive tract. These then leave the digestive tract and synthesize their way to the lungs where they develop into L4 larvae and, finally, the L4 re-enter the digest tract and move to the small intestine where they develop into adulthood. The adults remain in this anaerobic environment for the remainder of their life. The free-living larval stages have relatively low RQ (~35% of total quinones), whereas adults have high RQ ( ~100% of total quinones) (*Takamiya et al., 1993*). We used RNA-seq data (*Wang et al., 2011*; *Wang et al., 2012*) to analyze *coq-2* isoforms in free-living stages and in the adults to examine whether there was a switch from *coq-2a* to *coq-2e* as the parasites switch from low RQ aerobic-respiring embryos and larvae to high

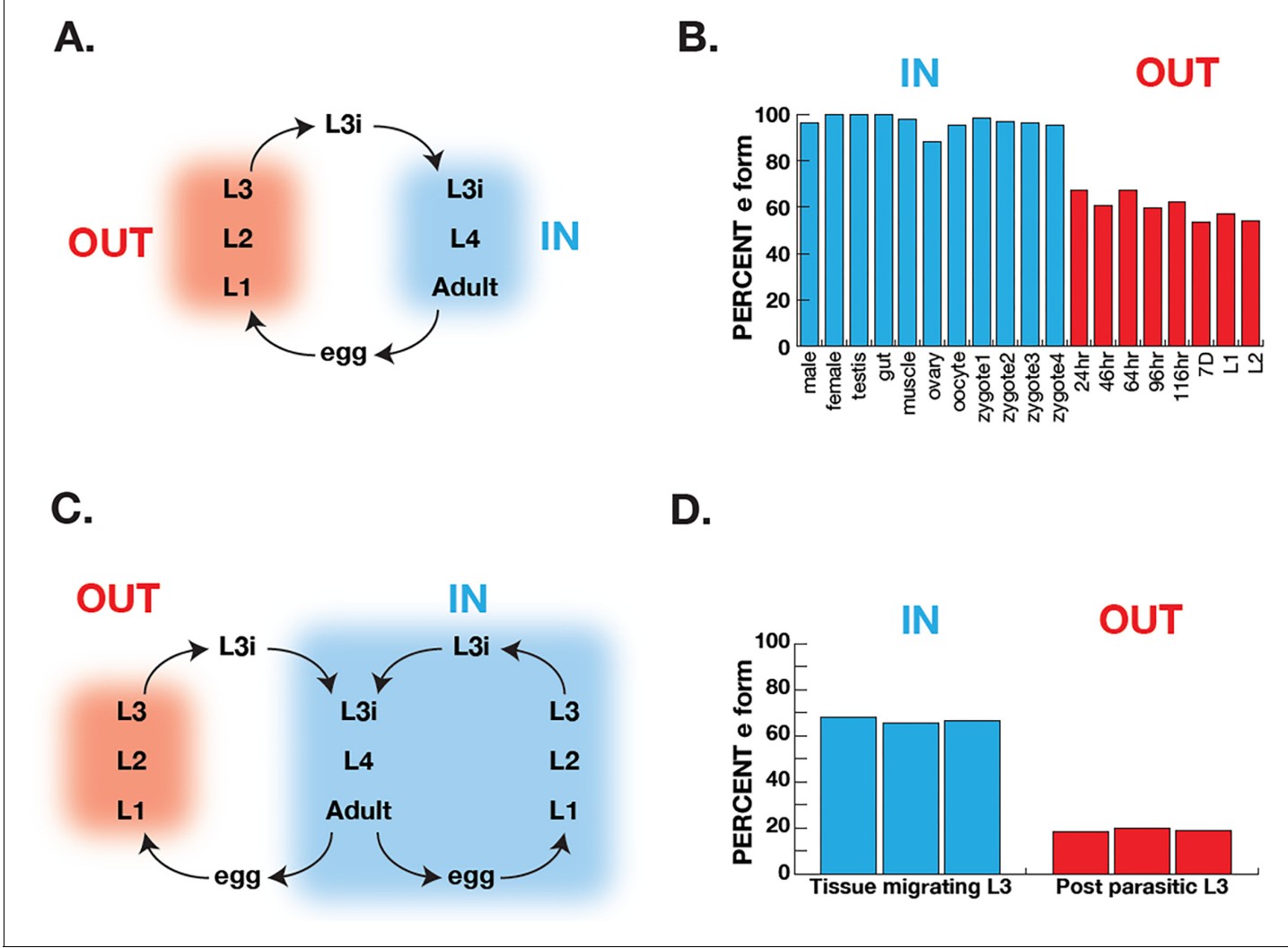

**Figure 6.** Correlation of COQ-2 splicing with change from aerobic to anaerobic life stages. (**A**) Schematic of the life cycle of *A. suum*. 'OUT' denotes aerobically respiring free-living stages; 'IN' indicates stages living inside the host intestine. (**B**) Graph indicates the percentage of all COQ-2 transcripts that include the RQ-specific exon (Percent e-form) in a number of life cycle stages, sexes, and tissues. Timing of embryogenesis shows the post-fertilization time in hours. (**C**) Schematic of the life cycle of *Strongyloides stercoralis*. 'OUT' denotes aerobically respiring free-living stages; 'IN' indicates stages living inside the host. Note that egg, L1, L2, and L3 can either develop inside or outside the host. (**D**) Graph indicates the percentage of all COQ-2 transcripts that include the RQ-specific exon (Percent e-form) in L3 larvae that either developed outside the host (OUT) or inside the host (IN). Data were derived from three individual replicates taken from published RNA-seq data (*Stoltzfus et al., 2012*).

RQ anaerobic adults. We see a clear switch approximately 60% of *coq-2* transcripts are *coq-2e* in the free-living, aerobic stages but >90% is the RQ-synthesizing *coq-2e* form in adults (*Figure 6B*). Increased *coq-2e* levels thus correlate with increased RQ levels.

The analysis in *Ascaris* is complicated by the life cycle: while we are comparing *coq-2* splicing in two distinct environments, this is also necessarily a comparison between different life stages (we are comparing developing embryos or larvae with adults). It is possible that the changes in splicing we see are not environmentally-induced by the switch from normoxia to anaerobic conditions but are simply a developmentally programmed switch. To address this, we turned to *Strongyloides stercoralis*. The life cycle of *S. stercoralis* (schematic *Figure 6C*) is broadly similar to *Ascaris* — L1-L3 stages are free-living, the L3 infective stage infects hosts, and L4 larvae and adults develop and live in the host. However, they have an alternative life cycle where instead of L1-L3 developing outside the host, the eggs can hatch in the host and the entire life cycle takes place inside the host. This allows

us to compare the same larval stage in two conditions — here we compare L3 animals that developed inside the host anaerobic environment with L3 animals that developed outside the host. The difference is clear about <20% of *coq-2* transcripts are *coq-2e* in the free-living L3s but >60% is the RQ-synthesizing *coq-2e* form in L3s that developed inside the host (*Figure 6D*; *Stoltzfus et al., 2012*). We conclude that increased inclusion of the *coq-2e*-specific exon correlates with increased synthesis of RQ during the parasite life cycles.

In summary, our data show that animal species that synthesize RQ regulate the choice between synthesizing UQ and RQ by alternative splicing of the polyprenyltransferase *COQ-2 gene*. A switch between two mutually exclusive exons changes the core of the COQ-2 enzyme and switches it from primarily generating UQ precursors to primarily RQ precursors. We propose that this switch ultimately results in a shift in quinone content from high UQ in aerobic conditions to high RQ in anaerobic conditions. This alternative splicing event is only seen in species that synthesize RQ and the switch correlates with the change from aerobic to anaerobic metabolism in parasitic helminths.

## Discussion

Organisms are continually challenged by changes in their environments and they must be able to respond for them to survive. Hypoxia is one such challenge and animals have evolved diverse strategies to alter their metabolism to cope with low oxygen levels. For example, humans rapidly switch to anaerobic glycolysis, generating lactate; goldfish on the other hand can adapt to hypoxia by fermenting carbohydrates to generate ethanol (*Shoubridge and Hochachka, 1980*). In this paper, we focus on how helminths can survive in anaerobic conditions by switching from ubiquinone (UQ)-dependent aerobic metabolism to rhodoquinone (RQ)-dependent anaerobic metabolism.

The ability to use RQ is highly restricted among animal species — only helminths, mollusks, and annelids are known to synthesize and use RQ. This is a key adaptation since it allows them to rewire their mitochondrial electron transport chain (ETC) to use a variety of terminal electron acceptors in the place of oxygen. They can, therefore, still use Complex I to pump protons and generate the proton motive force need to power the F0F1-ATPase in the absence of oxygen. This allows them to survive without oxygen for long periods — they are thus facultative anaerobes. *C. elegans* is a free-living nematode that does not face extended anaerobic conditions as an obligate part of its life cycle. However, *Pseudomonas aeruginosa*, a natural pathogen of *C. elegans*, uses cyanide to kill the worm (*Gallagher and Manoil, 2001*). Cyanide blocks the conventional ETC at Complex IV preventing the use of oxygen as a final electron acceptor. We speculate that in *C. elegans,* the alternative ETC may be an adaptive strategy to withstand transient cyanide stress from pathogens. For parasitic helminths like *Ascaris*, the need to respire anaerobically is critical for their life cycle since they must survive for long periods in the anaerobic environment of the human gut. Since the host does not synthesize RQ or use RQ-dependent metabolism if we could interfere with RQ synthesis, this would be an excellent way to target the parasite and leave the host untouched.

We previously showed that the key decision on whether to synthesize UQ to power aerobic metabolism or RQ to synthesize anaerobic metabolism is dictated by the choice of substrate of the polyprenyltransferase COQ-2 (*Del Borrello et al., 2019*; *Roberts Buceta et al., 2019*). COQ-2 must switch from using 4HB to synthesize UQ in aerobic conditions to 3HA to synthesize RQ in anaerobic conditions. Here we reveal the simple mechanism for that switch in substrate specificity in helminths: they use the mutually exclusive alternative splicing of two internal exons to remodel the core of COQ-2. Inclusion of exon 6a results in the COQ-2a enzyme that can synthesize UQ but not RQ; switching exon 6a for the alternative exon 6e yields COQ-2e which principally synthesizes RQ (*Figure 7*). All eukaryotes synthesize a homolog of COQ-2a — only the species known to synthesize RQ (helminths, annelids, and mollusks) have genomes encoding the RQ-specific exon 6e and this exon is introduced by alternative splicing in a similar mutually exclusive splicing event in all these phyla. These different lineages thus have the same solution to the problem of substrate switching in COQ-2 — to have two distinct forms of COQ-2 due to alternative splicing — and all do it with the same structural switch in COQ-2.

The alternative splicing of COQ-2 in all animal lineages that synthesize RQ draws focus to COQ-2 as a potential anthelmintic target. COQ-2e is required for RQ generation and has a distinct sequence to the pan-eukaryotic COQ-2a. The switch between COQ-2a to COQ-2e causes a change in the core of the COQ-2 active site and all species that synthesize RQ have a pair of conserved

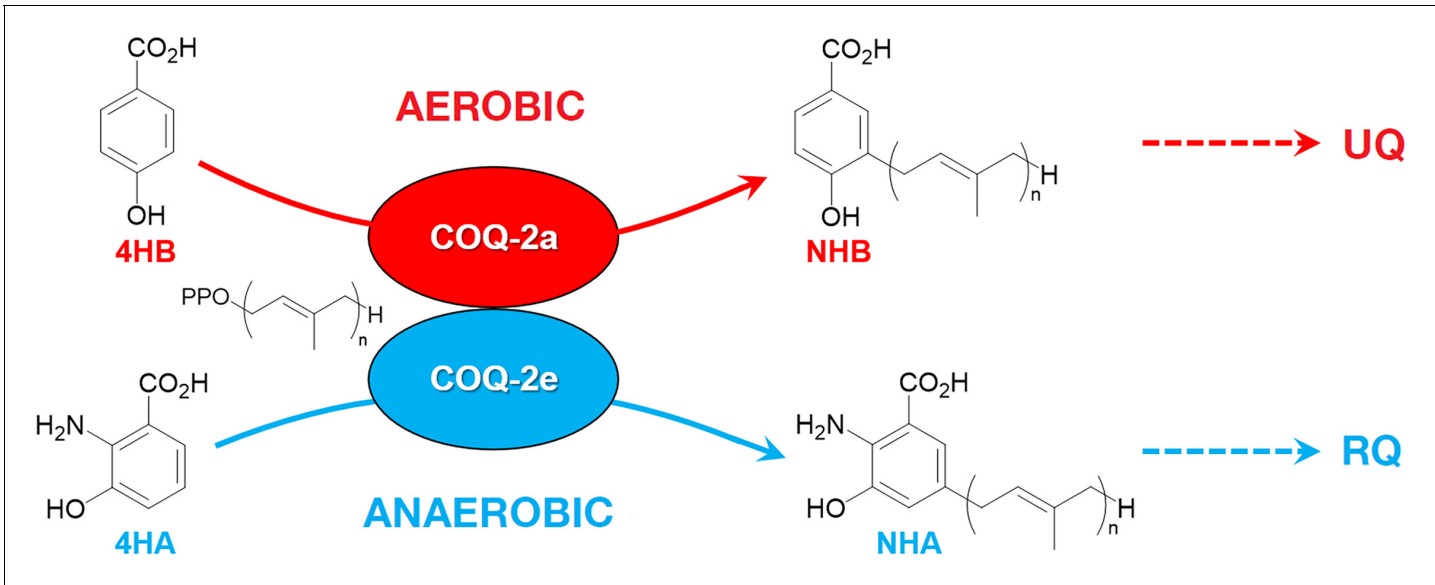

**Figure 7.** Discrimination between RQ and UQ biosynthesis in *C. elegans*. There are two variants of exon 6 in the *C. elegans coq-2* gene (6a and 6e) which undergo mutually exclusive alternative splicing leading to COQ-2a and COQ-2e isoforms, respectively. Synthesis of UQ originates from 4-hydroxybenzoic acid (4HB) and prenylation is catalyzed by COQ-2a (and marginally by COQ-2e) to form 4-hydroxy-3-nonaprenylbenzoic acid (NHB). By contrast, RQ is most likely synthesized from 3HA, and prenylation is facilitated by COQ-2e to form NHA. Several additional steps are required to convert NHB to UQ and NHA to RQ, respectively. Supplementary Documentation.

residues in COQ-2e. We suggest that small molecule inhibitors that selectively target COQ-2e and not the host COQ-2 enzyme could be potent anthelmintics and our research groups are initiating these screens at this point. We also note that it is rare to see such a clear and profound change in enzyme specificity due to a single splice event affecting the core of an enzyme catalytic site. There are other examples, most notably a switch in the substrate specificity of the cytochrome P450 CYP4F3 from LTB4 to arachidonic acid (*Christmas et al., 1999*; *Christmas et al., 2001*). This switch is tissue specific rather than environmentally induced, however, and there are few other examples to our knowledge. The regulation of COQ-2 specificity by alternative splicing is thus a beautiful and rare example of this type of enzyme regulation by alternative splicing. We note that while alternative splicing of COQ-2 appears to be the key regulated step in determining RQ or UQ biosynthesis, we see no such splicing regulation for other enzymes that are required for both RQ and UQ synthesis downstream of COQ-2 (*Roberts Buceta et al., 2019*), for example, there are no known splice variants of COQ-3 and COQ-5, which are quinone methylases downstream of COQ-2. This suggests that while COQ-2 can clearly discriminate between substrates that have/lack a 2-amino group, COQ-3 and COQ-5 would be more promiscuous than COQ-2 and act on both RQ and UQ precursors.

Finally, the mechanism of alternative splicing to regulate the synthesis of UQ or RQ may explain some part of the evolution of RQ synthesis in animals and its phylogenetic distribution. One of the enduring mysteries of RQ synthesis is that it only occurs in very restricted animal phyla. RQ synthesis could either be the ancestral state with widespread loss across most animal species or else RQ synthesis could have arisen independently in helminths, annelids, and mollusks. It is unclear that our findings here definitively answer this but we do note that there is a precedent for a mutually exclusive splicing event that has arisen independently in helminths, annelids, and mollusks in the gene *mrp-1* (*Yue et al., 2017*) which encodes a transporter that is required for the uptake of vitamin B12 into mitochondria. The *mrp-1* gene undergoes alternative splicing in these species and just like *coq-2*, *mrp-1* has mutually exclusive exons of exactly the same length. Importantly, in *mrp-1*, a mutually exclusive alternative splicing has arisen independently in each lineage in which it is seen. This example indicates it is at least plausible that alternative splicing of *coq-2* might have originated independently in helminths, annelids, and mollusks, but we do not believe that our current data can distinguish between an ancestral animal origin of RQ biosynthesis followed by independent losses,

or independent animal origin of RQ biosynthesis due to an '*mrp-1*-like' independent evolution of alternative splicing of *coq-2*. Future studies should shed light on this issue. Whatever the evolutionary scenario, our results show that a simple innovation of a mutually exclusive alternative splicing event can have a profound effect rewiring animal metabolism and it will be interesting to establish whether alternative splicing of COQ-2 is sufficient to drive RQ biosynthesis, or whether additional innovations are needed.

# Materials and methods

## Key resources table

| Reagent type (species) or resource | Designation | Source or reference | Identifiers | Additional information |
|---|---|---|---|---|
| Gene (*Caenorhabditis elegans*) | *coq-2* | WormBase | WBGene00000762 | |
| Strain, strain background (*Caenorhabditis elegans*) | N2 | *Caenorhabditis* Genetics Center (CGC) | N2 | Wild-type |
| Strain, strain background (*Caenorhabditis elegans*) | *coq-2 (syb1715)* | This paper | PHX1715 | *coq-2Δ6a* **Supplementary file 1** |
| Strain, strain background (*Caenorhabditis elegans*) | *coq-2 (syb1721)* | This paper | PHX1721 | *coq-2Δ6e* **Supplementary file 1** |
| Strain, strain background (*Escherichia coli*) | OP50 | *Caenorhabditis* Genetics Center (CGC) | OP50 | |
| Sequence-based reagent | sgRNAs used in this study | This paper | | **Supplementary file 1** |
| Sequence-based reagent | PCR primers used in this study | This paper | | **Supplementary file 1** |
| Chemical compound, drug | Potassium cyanide (KCN) | Sigma-Aldrich | 60178–25G | Stock solution: 50 mM in PBS buffer |
| Other | Hexane, HPLC grade | Sigma-Aldrich | 650552–1L | |
| Other | Acetonitrile, LC-MS grade | Fisher Scientific | A955-4 | |
| Other | Ubiquinone-3 standard | **Campbell et al., 2019** | | |
| Other | Ubiquinone-9 standard | Sigma-Aldrich | 27597–1 MG | |
| Other | Rhodoquinone-9 standard | **Roberts Buceta et al., 2019** | | Isolated from *Ascaris suum* |
| Software, algorithm | Whippet | **Sterne-Weiler et al., 2018** | RRID:SCR_018349 | https://github.com/timbitz/Whippet.jl |
| Software, algorithm | HISAT2 | **Kim et al., 2019** | RRID:SCR_015530 | https://daehwankimlab.github.io/hisat2/ |
| Software, algorithm | UCSF Chimera | Resource for Biocomputing Visualization and Informatics **Pettersen et al., 2004** | RRID:SCR_004097 | http://plato.cgl.ucsf.edu/chimera/ |
| Software, algorithm | MODELLER | University of California at San Francisco **Webb and Sali, 2016** | RRID:SCR_008395 | https://salilab.org/modeller/ |
| Software, algorithm | Image analysis pipeline | **Spensley et al., 2018** | | https://github.com/fraser-lab-UofT/acute_assay |
| Software, algorithm | Python | Python | RRID:SCR_008394 | https://www.python.org/ |
| Software, algorithm | NIS-Elements | Nikon | RRID:SCR_014329 | https://www.microscope.healthcare.nikon.com/products/software/nis-elements |

*Continued on next page*

*Continued*

| Reagent type (species) or resource | Designation | Source or reference | Identifiers | Additional information |
|---|---|---|---|---|
| Software, algorithm | ImageMagick | ImageMagick | RRID:SCR_014491 | https://imagemagick.org/ |
| Software, algorithm | BioFormats | OME - Open Microscopy Environment | RRID:SCR_000450 | https://docs.openmicroscopy.org/bio-formats/5.7.1/users/comline tools/index.html |

## Sequence identification and analysis

To identify COQ-2 sequences from lineages known to synthesize RQ, we searched and analyzed from genomes and transcriptomes of platyhelminths (*Schistosoma mansoni*, *Fasciola hepatica,* and *Schmidtea mediterranea*), nematodes (*Ascaris suum*, *Brugia malayi*, *Haemonchus contortus*, *Trichuris muris*, *Strongyloides stercoralis*, *Strongyloides ratti*, *Ancylostoma ceylanicum,* and *C. elegans*), mollusks (*Crassostrea virginica,* and *Biomphalaria glabrata*), and annelids (*Capitella teleta*). Sequences were retrieved from https://parasite.wormbase.org (WBPS14), *S. mediterranea* database (http://smedgd.neuro.utah.edu), NCBI protein and nucleotide databases and UniProt (https://www.uniprot.org). Human, mice, and *Saccharomyces cerevisiae* COQ-2, eukaryotic lineages are known to be unable to synthesize RQ, were also identified for comparison. Searches were performed initially with BLASTP (protein databases) using human and *C. elegans* COQ-2 sequences as queries. Additionally, TBLASTN searches were performed using genomic sequences and cDNAs databases. This served to confirm the annotated protein sequences and to identify the non-annotated ones. Identified sequences were confirmed by best reciprocal hits in BLAST. Multiple sequence alignments were made with MUSCLE 3.8 (*Chojnacki et al., 2017*). Gaps were manually refined after alignment inspection.

## RNA-seq analysis of mutually exclusive exons

To confirm if *coq-2* exons are spliced in a mutually exclusive manner in other helminths, mollusks and annelids, we analyzed the existing RNA-seq data for evidence of alternative splicing (listed in *Supplementary file 3*). Whippet (v0.11) (*Sterne-Weiler et al., 2018*) was used to analyze RNA-seq data for quantification of AS events. To create a splicing index of exon-exon junctions in Whippet, genome annotations were taken from WormBase Parasite (WBPS14) and Ensembl Metazoa (Release 45). To identify novel exons and splice sites, reads were first aligned to the genome using HISAT2 (*Kim et al., 2015*). The BAM file generated was then used to supplement the existing genome annotations to create a splicing index of known and predicted exon-exon junctions in Whippet (using the −bam −bam-both-novel settings). Where required, TBLASTN data was also used to guide manual re-annotation of the *coq-2* gene. Quantification of AS events was then performed by running whippet quant at default settings. This analysis was repeated for all species listed in *Figure 5* using publicly available RNA-seq datasets (details on datasets used can be found in *Supplementary file 3*). A summary of *coq-2* exons with reads that mapped to alternative exon-exon junctions are listed in *Supplementary file 3*. We also identified cases where both *coq-2* exons were either included or skipped. However, since these are likely to be non-productive transcripts due to a pre-mature termination codon, we expressed exon usage as the proportion of events where either only the 'a' or the 'e' form is included.

## Structural analysis

Multiple sequence alignment was performed using Clustal Omega (*Madeira et al., 2019*). The substrate-bound structure of a UbiA homolog from *A. pernix* (PDB: 4OD5) was displayed on Chimera (*Pettersen et al., 2004*) and the *C. elegans* sequence was threaded by homology using Modeller (*Sali and Blundell, 1993*; *Webb and Sali, 2016*).

## *Caenorhabditis elegans* strains and culture conditions

The *C. elegans* wild-type Bristol strain (N2) was obtained from the Caenorhabditis Genetics Center (CGC, University of Minnesota, USA), which is supported by the National Institutes of Health-Office of Research Infrastructure Programs. The *C. elegans* mutant strains in *coq-2* exon 6A (PHX1715, *coq-2(syb1715)*) and *coq-2* exon 6E (PHX1721, *coq-2(syb1721)*) were generated by Suny Biotech Co., Ltd

(Fuzhou City, China) using CRISPR/Cas9 system. The precise deletion of both mutant strains (134 bp) was verified by DNA sequencing the flanking region of exons 6a and 6e. The wild-type sequence, the deleted sequence in each strain, the sgRNAs, and primers used are listed in *Supplementary file 1*.

The general methods used for culturing and maintenance of *C. elegans* are described in *Brenner, 1974*. All chemical reagents were purchased from Sigma-Aldrich (St. Louis, MO). The *E. coli* OP50 strain, used as *C. elegans* food, was also received from CGC.

### Lipid extraction and LC-MS quantitation

Lipid extractions of *C. elegans* N2 and mutant strains were performed on ~100 mg worm pellets after adding 1000 pmol $UQ_3$ internal standard (*Roberts Buceta et al., 2019*). LC-MS samples were prepared from lipid extracts and diluted 1:100 (*Bernert et al., 2019*). Standards were extracted using the same method as for worm samples at the following concentrations: $UQ_3$ (10 pmol/10 μL injection), $RQ_9$ (0.75, 1.5, 3.0, 4.5, or 6.0 pmol/10 μL injection), and $UQ_9$ (3.75, 7.5, 15.0, 22.5, or 30.0 pmol/10 μL injection). The $UQ_3$ standard was synthesized at Gonzaga University (*Campbell et al., 2019*), the $RQ_9$ standard was isolated by preparative chromatography from *A. suum* lipid extracts (*Roberts Buceta et al., 2019*) and the $UQ_9$ standard was purchased (Sigma-Aldrich, St. Louis, MO). The general LC-MS conditions and parameters were previously reported (*Bernert et al., 2019*; *Campbell et al., 2019*). Samples were analyzed in quadruplicate and the pmol quinone was determined from the standard curve and corrected for recovery of internal standard. Samples were normalized by mg pellet mass.

### Image-based KCN recovery assay

The KCN recovery assay was performed as previously described (*Del Borrello et al., 2019*; *Spensley et al., 2018*). Briefly, L1 worms were isolated by filtration through an 11 μm nylon mesh filter (Millipore: S5EJ008M04). Approximately, 100 L1 worms in M9 were dispensed to each well of a 96-well plate and an equal volume of potassium cyanide (KCN) (Sigma-Aldrich, St. Louis, MO) solution was then added to a final concentration of 200 μM KCN. Upon KCN addition, the plates were immediately sealed and incubated at room temperature for 15 hr on a rocking platform. After 15 hr, KCN was diluted 6-fold by addition of M9 buffer. Plates were immediately imaged on a Nikon Ti Eclipse microscope every 10 min for 3 hr. Fractional mobility scores (FMS) were then calculated using a custom image analysis pipeline (*Spensley et al., 2018*). For each strain, FMSfor the KCN-treated wells were normalized to the M9-only control wells at the first timepoint. Three technical replicates were carried out in each experiment and the final FMS taken from the mean of four biological replicates.

## Acknowledgements

*C. elegans* strain N2 and *E. coli* OP50 were provided by the Caenorhabditis Genetics Center, which is funded by NIH Office of Research Infrastructure Program: P40 OD010440. We thank Exequiel Barrera, John Calarco, and Amy Caudy for helpful discussions. We would also like to thank Amy Buck, Niki Gounaris, Jim Collins, Rick Maizels, and Murray Selkirk for organizing the Hydra meeting on Parasitic Helminths where this collaboration started.

## Additional information

### Funding

| Funder | Grant reference number | Author |
| --- | --- | --- |
| Canadian Institutes of Health Research | 501584 | Andrew G Fraser |
| Canadian Institutes of Health Research | 5003009 | Andrew G Fraser |
| Agencia Nacional de Investigación e Innovación | FCE_2014_1_104366 | Gustavo Salinas |

| Agencia Nacional de Investigación e Innovación | FCE_1_2019_1_155779 | Gustavo Salinas |

The funders had no role in study design, data collection and interpretation, or the decision to submit the work for publication.

## Author contributions

June H Tan, Data curation, Formal analysis, Investigation, Writing - review and editing; Margot Lautens, Data curation, Formal analysis, Investigation, Visualization; Laura Romanelli-Cedrez, Jianbin Wang, Data curation, Formal analysis, Investigation; Michael R Schertzberg, Conceptualization, Data curation, Investigation; Samantha R Reinl, Formal analysis, Investigation; Richard E Davis, Supervision, Funding acquisition, Investigation, Project administration; Jennifer N Shepherd, Gustavo Salinas, Conceptualization, Supervision, Funding acquisition, Investigation, Writing - original draft, Project administration, Writing - review and editing; Andrew G Fraser, Conceptualization, Supervision, Funding acquisition, Writing - original draft, Project administration, Writing - review and editing

## Author ORCIDs

June H Tan (iD) http://orcid.org/0000-0001-6597-3952
Margot Lautens (iD) http://orcid.org/0000-0002-8503-9603
Andrew G Fraser (iD) https://orcid.org/0000-0001-9939-6014

## Decision letter and Author response

Decision letter https://doi.org/10.7554/eLife.56376.sa1
Author response https://doi.org/10.7554/eLife.56376.sa2

# Additional files

## Supplementary files

- Supplementary file 1. *C. elegans* strains.
- Supplementary file 2. Statistical analysis of $RQ_9$ and $UQ_9$ levels in *coq-2* mutant strains.
- Supplementary file 3. Genomic coordinates of known and predicted mutually exclusive *coq-2* a/e exons.
- Transparent reporting form

## Data availability

All data generated or analysed during this study are included in the manuscript and supporting files.

The following previously published datasets were used:

| Author(s) | Year | Dataset title | Dataset URL | Database and Identifier |
|---|---|---|---|---|
| Gao F, Liu X, Wu XP, Wang XL, Gong D, Lu H, Xia Y, Song Y, Wang J, Du J, Liu S, Han X, Tang Y, Yang H, Jin Q, Zhang X, Liu M | 2012 | Differential methylation in discrete developmental stages of the parasitic nematode Trichinella spiralis | https://www.ebi.ac.uk/ena/data/view/PRJNA170655 | EBI European Nucleotide Archive, PRJNA170655 |
| Foth BJ, Tsai IJ, Reid AJ, Bancroft AJ, Nichol S, Tracey A, Holroyd N, Cotton JA, Stanley EJ, Zarowiecki M, Liu JZ, Huckvale T, Cooper PJ, Grencis RK, Berriman M | 2014 | RNA-seq of Trichuris muris in different developmental stages | https://www.ebi.ac.uk/ena/data/view/PRJEB1054 | EBI European Nucleotide Archive, PRJEB1054 |
| Wang J, Czech B, | 2011 | Ascaris suum transcriptome (RNA- | https://www.ebi.ac.uk/ | EBI European |

| | | | | |
|---|---|---|---|---|
| Crunk A, Wallace A, Mitreva M, Hannon GJ, Davis RE | | Seq) data | ena/data/view/ PRJNA142041 | Nucleotide Archive, PRJNA142041 |
| Wang J, Mitreva M, Berriman M, Thorne A, Magrini V, Koutsovoulos G, Kumar S, Blaxter ML, Davis RE | 2012 | Silencing of Germline-Expressed Genes by DNA Elimination in Somatic Cells | http://www.ebi.ac.uk/ ena/data/view/ SRP013573 | EBI European Nucleotide Archive, SRP013573 |
| Choi YJ, Ghedin E, Berriman M, McQuillan J, Holroyd N, Mayhew GF, Christensen BM, Michalski ML | 2011 | Transcription profiling of of the human filarial nematode Brugia malayi across multiple life-cycle stages | http://www.ebi.ac.uk/ ena/data/view/ ERP000948 | EBI European Nucleotide Archive, ERP000948 |
| The Wellcome Trust Sanger Institute | 2013 | RNA-seq of Strongyloides ratti to investigate molecular basis of parasitism | http://www.ebi.ac.uk/ ena/data/view/ ERP002187 | EBI European Nucleotide Archive, ERP002187 |
| Stoltzfus JD, Minot S, Berriman M, Nolan TJ, Lok JB | 2012 | RNAseq of S. stercoralis PV001 strain developmental stages | http://www.ebi.ac.uk/ ena/data/view/ ERP001556 | EBI European Nucleotide Archive, ERP001556 |
| Baskaran P, Jaleta TG, Streit A, Rödelsperger C | 2017 | Duplications and positive selection drive the evolution of parasitism associated gene families in the nematode Strongyloides papillosus | https://www.ebi.ac.uk/ ena/data/view/ PRJEB14543 | EBI European Nucleotide Archive, PRJEB14543 |
| The Genome Institute | 2016 | Ancylostoma ceylanicum Genome Sequencing | https://www.ebi.ac.uk/ ena/data/view/ PRJNA72583 | EBI European Nucleotide Archive, PRJNA72583 |
| Laing R, Kikuchi T, Martinelli A, Tsai IJ, Beech RN, Redman E, Holroyd N, Bartley DJ, Beasley H, Britton C, Curran D, Devaney E, Gilabert A, Hunt M, Jackson F, Johnston SL, Kryukov I, Li K, Morrison AA, Reid AJ, Sargison N, Saunders GI, Wasmuth JD, Wolstenholme A, Berriman M, Gilleard JS, Cotton JA | 2013 | Haemonchus contortus transcriptome | http://www.ebi.ac.uk/ ena/data/view/ ERP002173 | EBI European Nucleotide Archive, ERP002173 |
| Schwarz EM, Korhonen PK, Campbell BE, Young ND, Jex AR, Jabbar A, Hall RS, Mondal A, Howe AC, Pell J, Hofmann A, Boag PR, Zhu XQ, Gregory T, Loukas A, Williams BA, Antoshechkin I, Brown C, Sternberg PW, Gasser RB | 2013 | Haemonchus contortus transcriptome (from Haecon-5) | https://www.ebi.ac.uk/ ena/data/view/ PRJNA205196 | EBI European Nucleotide Archive, PRJNA205196 |
| Eccles D, Chandler J, Camberis M, Henrissat B, Koren S, Le Gros G, Ewbank JJ | 2018 | Nippostrongylus brasiliensis Genome & Transcriptome sequencing and assembly | https://www.ebi.ac.uk/ ena/data/view/ PRJEB20824 | EBI European Nucleotide Archive, PRJEB20824 |
| Cwiklinski K, Dalton JP, Dufresne PJ, La Course J, Williams DJ, Hodgkinson J, Paterson S | 2015 | RNAseq analysis of the liver fluke Fasciola hepatica | https://www.ebi.ac.uk/ ena/data/view/ ERR576952 | EBI European Nucleotide Archive, ERR576952 |

| | | | | |
|---|---|---|---|---|
| Protasio AV, Dunne DW, Berriman M | 2013 | RNA-Seq of Schistosoma mansoni (flatworms) larva and adult individuals at different life-stages | http://www.ebi.ac.uk/ena/data/view/ERP000427 | EBI European Nucleotide Archive, ERP000427 |
| Zhang G, Fang X, Guo X, Li L, Luo R, Xu F, Yang P, Zhang L, Wang X, Qi H, Xiong Z, Que H, Xie Y, Holland PW, Paps J, Zhu Y, Wu F, Chen Y, Wang J, Peng C, Meng J, Yang L, Liu J, Wen B, Zhang N, Huang Z, Zhu Q, Feng Y, Mount A, Hedgecock D, Xu Z, Liu Y, Domazet-Lošo T, Du Y, Sun X, Zhang S, Liu B, Cheng P, Jiang X, Li J, Fan D, Wang W, Fu W, Wang T, Wang B, Zhang J, Peng Z, Li Y, Li N, Chen M, He Y, Tan F, Song X, Zheng Q, Huang R, Yang H, Du X, Chen L, Yang M, Gaffney PM, Wang S, Luo L, She Z, Ming Y, Huang W, Huang B, Zhang Y, Qu T, Ni P, Miao G, Wang Q, Steinberg CE, Wang H, Qian L, Zhang G, Liu X, Yin Y | 2012 | RNA-seq of Pacific oyster Crassostrea gigas under different developmental stages and stress treatments | http://www.ebi.ac.uk/ena/data/view/SRP014559 | EBI European Nucleotide Archive, SRP014559 |
| Kenny NJ, Truchado-García M, Grande C | 2016 | Biomphalaria glabrata Transcriptome or Gene expression | https://www.ebi.ac.uk/ena/data/view/PRJNA306682 | EBI European Nucleotide Archive, PRJNA306682 |
| Burns R, Pechenik J | 2016 | Capitella teleta RNA sequencing of competent and pre-competent larvae | https://www.ebi.ac.uk/ena/data/view/PRJNA379706 | EBI European Nucleotide Archive, PRJNA379706 |

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
