## [Decision Letter]

Thank you for submitting your article "Alternative splicing of COQ-2 determines the choice between ubiquinone and rhodoquinone biosynthesis in animals" for consideration by *eLife*. Your article has been reviewed by three peer reviewers, and the evaluation has been overseen by Gisela Storz as Reviewing and Senior Editor. The reviewers have opted to remain anonymous.

The reviewers have discussed the reviews with one another and the Reviewing Editor has drafted this decision to help you prepare a revised submission.

Summary:

Parasitic helminths are important human pathogens. It is important to find candidate therapies that specifically target these pathogens. Many parasitic helminths use rhodoquinone (RQ) instead of ubiquinone (UQ) when they live inside a host, in an anaerobic environment. Since the host does not make or use RQ, it could be an important pharmacological target. The authors previously showed that switch from UQ to RQ synthesis in parasitic helminths is driven by a change of substrate specificity of the polyprenyltransferase COQ-2 (Del Borrello et al., 2019; Roberts Buceta et al., 2019) when their living environment is switched from normoxia to anaerobic conditions. In this paper, they answer a simple but important question: what is the mechanism for substrate switching for COQ-2? The authors found that coq-2 has two different splicing isoforms, coq-2a and coq-2e, and that COQ-2a is specific for UQ synthesis while COQ-2e is specific for RQ synthesis. They use CRISPR/Cas9 to specifically delete each isoform in *C. elegans* and then measure UQ and RQ levels. The results clearly demonstrate the specificity of COQ-2a and COQ-2e for UQ and RQ, respectively. They further show the functional significance of this as animals that cannot synthesize RQ are hypersensitive to cyanide, which blocks the electron transport chain, and is a proxy for anaerobic conditions.

The proposed model is attractive and seems very promising and overall the reviewers were enthusiastic about the study. However, the paper is light on results. After discussion, the reviewers recommend that you consider the following experiments:

1) Directly validate splice forms' abundance on the RNA levels. The RNA-seq data meta-analysis is reasonable, but it would be good to see this validated.

2) Measure the level of 4HB and 3HA. If only the splice form selection drives the switch in quinone type then 4HB and 3HA levels should not change drastically.

3) Titrate KCN in Figure 5C experiment and also feed the helminths with 3HA. Data show that even the UQ-specific form can make a small amount of RQ. Is it enough to survive lower KCN treatment and can it be boosted by 3HA feeding?

4) Can the e-exon, which switches COQ2 to make more RQ, be found in any other organism? It was not clear in the text and methods if the sequence was "BLASTed" against all organisms to see if the e-exon exists in any non-Nematodes.

5) Analyze the in silico model and do binding simulations to explain why F204L and A243S changes would so strongly dictate precursor selection. In the human system, COQ2 can use many modified headgroups so why two mutations make it so specific?

Editorial points:

The text should also be revised to address the following:

– Throughout: The text could be improved to correctly describe the results of the experiments. For example, in the final conclusion authors say that "A switch between two mutually exclusive exons changes the core of the COQ-2 enzyme and switches it from generating UQ precursors to RQ precursors" however, Figure 5C shows that RQ-specific form makes both RQ and UQ, and also UQ-specific form can make a small amount of RQ. Therefore, it is somewhat misleading to suggest that individual splice forms can make only UQ or RQ.

– Introduction paragraph three: Maybe just start sentence with "RQ biosynthesis in animals requires precursors derived from tryptophan" (the evidence in Stairs is not convincing, it could be bacteria in the medium), the situation in microbes does not bear upon these finding. Bernet et al. is convincing, but concerns the R. rubrum gene.

– Please explain the conversion of the amino group in PABA to the hydroxyl group in HHB in Figure 2C.

– Introduction paragraph four: summarize the findings and restate the Abstract (mollusc and mollusk spelling occur). The meaning of "independent evolution" in is unclear.

– Figure 1 legend: ETF is not shown in the figure. For malate dismutation to operate (which it does, the endproduct stoichiometry is correct), fumarate reductase (CII) is the RQ oxidant (Müller et al., MMBR 2012).

– Subsection “Regulation of the alternative splicing of *coq-2* in helminths” paragraph four: the data show that animal species

– Discussion paragraph three: all animal lineages

– Discussion: The authors favour independent origin of this mechanism and the exons. One needs to spell out what that means. That would mean that sequence homologous exons arose independently in independent lineages, which would mean that e arose from a each time, which is possible, and that e always arose in front of a in each of the 14 lineages surveyed, a 1/16000 proposition if all of the events were independent. Of course, if e arose from a in the common ancestor of these lineages followed by many differential losses in aerobic lineages, then we would have normal Darwinian evolution and no need for LGT or independent origins (googling differential gene loss in eukaryotes or evolution by gene loss returns a lot of hits, the evidence for the widespread occurrence of loss in eukaryotes is uncontroversial). Also possible. Of course, it is also possible that the alternative RQ specific exon here arose only once in one of these animals has been passed around via LGT among eukaryotic lineages (Leger et al., 2018) and specifically inserted into the COQ2 gene of the anaeobes in the conserved position (always in front of the UQ exon) in adaptation to anaerobic niches.

– What if we consider the possibility that a single origin and loss, not independent origins (by whatever mechanism) are the cause of the homologous exons (and exon order). What use would RQ be to the common ancestor of molluscs annelisds and nematodes? It would be essential.

– Animal aroses and diversified during a phase of Earth history in which oxygen was still low. Geologists have been saying for 20 years that the Proterozoic was anoxic (see Figure 1 in Lyons et al., 2014 or Figure 2 in Catling and Zahnle, 2020). Palaeontologists have been saying that animals arose and diversified in the Ediacaran (that is, Precambrian; <540 MY ago), nematode-mollusc divergence (do Reis et al., 2015) or the nematode annelid divergence (Parfrey et al., 2011). When I find 540 MY ago on the timescales in Lyons et al. and Catling and Zahnle, oxygen is low. Moreover, during the one billion years of eukaryote evolution before that, oxygen was low too. Some of us have been saying for 20 years (Zimorski et al., 2019; Gould et al., 2019) that survival in anaerobic environments was a normal thing that eukaryotes did for over a billlion years of evolution before life on land above the soli line. In that interpretation, the expectation would be that mechanisms of aerobic-anaerobic switching exist that are conserved across annelids, molluscs and nematodes (including free living and parasitic forms). My goodness, a Nobel Prize in Physiology and Medicine went for HIF last year, which senses low oxygen in animals. HIF is conserved across all animals. Did the HIF-dependent oxygen sensing cascade arise independently in all animals? Not likely. Was the HIF pathway laterally transferred? Not likely. I would think that the authors might wish to at least entertain the possibility that the anaerobic physiology that they see in these in animals (and eukaryotes) is conserved from the low oxygen and anaerobic past of animal and eukaryotic lineages. Is it possible that HIF sends the signal that regulates the RQ response? Yes, but I know of no evidence to suggest that, and there are HIF-independent possibilites. They mention Hochachka's work on vertebrates and the goldfish ethanol fermentation example. The vertebrate mechanisms (anaerobiosis in humans will work for about a minute) are not UQ dependent. This is the author's paper and they should write what they want. But if they say independent origins, that interpretation carries a lot of corollaries, none of which are spelled out in the paper, but are spelled out here. Of course, maybe 20 years of geochemical data on late oyxgen, the known age of fossil animals, the conservation of HIF and the conservation of pathways for eukaryote anaerobe survival are all wrong.

– Why does *C. elegans* make both RQ and UQ? A paragraph in the Discussion would be helpful, even if it is speculative.

– Do the other enzymes involved in RQ or UQ biosynthesis change in expression under anaerobic conditions?

[Editors' note: further revisions were suggested prior to acceptance, as described below.]

Thank you for submitting your article "Alternative splicing of COQ-2 determines the choice between ubiquinone and rhodoquinone biosynthesis in animals" for consideration by *eLife*. Your article has been reviewed by two peer reviewers, and the evaluation has been overseen by Gisela Storz as the Reviewing and Senior Editor. The reviewers have opted to remain anonymous.

Summary:

Parasitic helminths are important human pathogens. It is important to find candidate therapies that specifically target these pathogens. Many parasitic helminths use rhodoquinone (RQ) instead of ubiquinone (UQ) when they live inside a host, in an anaerobic environment. Since the host does not make or use RQ, it could be an important pharmacological target. The authors previously showed that switch from UQ to RQ synthesis in parasitic helminths is driven by a change of substrate specificity of the polyprenyltransferase COQ-2 (Del Borrello et al., 2019; Roberts Buceta et al., 2019) when their living environment is switched from normoxia to anaerobic conditions. In this paper, they answer a simple but important question: what is the mechanism for substrate switching for COQ-2? The authors found that coq-2 has two different splicing isoforms, coq-2a and coq-2e, and that COQ-2a is specific for UQ synthesis while COQ-2e is specific for RQ synthesis. They use CRISPR/Cas9 to specifically delete each isoform in *C. elegans* and then measure UQ and RQ levels. The results clearly demonstrate the specificity of COQ-2a and COQ-2e for UQ and RQ, respectively. They further show the functional significance of this as animals that cannot synthesize RQ are hypersensitive to cyanide, which blocks the electron transport chain, and is a proxy for anaerobic conditions.

Revisions:

Reviewer #2:

The authors have put worth a reasonable revision that provided very little new data, but nonetheless somewhat bolsters their compelling model. Given that the authors haven't, and perhaps can't, provide certain validations of their data, I think the title and some language should be toned down. In the end, both splice forms can make both quinones, but at a different ratio. Also, both splice forms are often co-expressed. The levels of 4HB and 3HA headgroups were not measured so how they impact levels of the quinones is unknown. It will require a significant amount of biochemical data to fully understand the exact differences between the COQ-2a and COQ-2e splice forms. However, it is quite unlikely that the general model put forth is incorrect.

Reviewer #3:

I continue to think this is a nice study that should be published in *eLife*. The authors did a nice job in their rebuttal and I am satisfied overall. I agree that RNA-seq is totally fine, given the depth and precision of the method. However, I find parts of the Discussion a bit unsatisfying. For instance, what is the function of alternative splicing of mrp-1? There is no reference for the statement that this transporter transports B12 into the mitochondria. Also, it is not clear that vitamin B12 is required for RQ synthesis in *C. elegans*, even though it is stated. Finally, the relationship with branched chain fatty acid synthesis is unclear to me. However, clarifying this is up to the authors and shouldn't prevent acceptance of the work.

---

## [Author Response]

Revisions for this paper:The proposed model is attractive and seems very promising and overall the reviewers were enthusiastic about the study. However, the paper is light on results. After discussion, the reviewers recommend that you consider the following experiments:1) Directly validate splice forms' abundance on the RNA levels. The RNA-seq data meta-analysis is reasonable, but it would be good to see this validated.

We agree with the reviewers that it is crucial that central methods are carefully validated. Our primary way to analyse splicing throughout this paper is with RNA-seq. We absolutely agree that such an approach must be validated carefully and we have done rigorous validation of the RNA-seq splicing analysis pipeline that we employ here, specifically in *C. elegans,* the model helminth. In our publication https://genome.cshlp.org/content/21/2/342.abstract, we used this same RNA-seq pipeline along with a completely independent method using microarrays to analyse splicing in *C. elegans* and extensively validated the data using RT-PCR. We found that the 2 independent platforms correlated with an R2=0.85 and furthermore, that we could confirm 92% of identified splice forms using RT-PCR. The RNA-seq pipeline is thus robust and accurate. In addition to that validation in our group, the *C. elegans* transcriptome has been also been analysed using direct sequencing on a nanopore platform (which we refer to in the text), another independent platform — the RNAseq data match very closely to the nanopore data for *coq-2*. Finally, another independent group specifically validated the *coq-2* mutually exclusive splicing event using RT-PCR ( https://pubmed.ncbi.nlm.nih.gov/25254147/ ). We therefore feel that our RNA-seq-based pipeline for analysing alternative splicing has been validated extremely carefully — nanopore, microarray, RT-PCR all confirm its accuracy and coverage in general and specifically in the case of *coq-2*, both nanopore and RT-PCR confirm our analysis.

This validation has all been done in *C. elegans*. We recognize that we did not validate the RNA-seq splicing analysis in the other species being discussed. But we are using exactly the same pipeline as has been extensively validated in *C. elegans*, a nematode, and are applying it to other nematodes. The data presented for *Ascaris* and *S. stercoralis* show extremely high agreement between independent biological repeats and these genomes are highly compact so splice analysis is easy and robust. We don’t fully understand why the reviewers think that our highly validated pipeline would be robust in one nematode but not in another — I’m unclear what the underlying hypothesis for this would be? It does feel that at some point a highly validated technology should be trusted unless there’s a clear reason for scepticism and I’m not sure what that would be in this case. At any rate, we are happy to include the following sentences in the paper:

“We note that while the alternative splicing of *coq-2* in *C. elegans* we observe using RNA-seq has been validated by direct sequencing, microarray analysis, and RT-PCR, the data on the other helminths only derives from RNA-seq and genome analysis. While it uses the same robust RNA-seq analysis pipeline, it would be ideal to validate these changes in the future.”

2) Measure the level of 4HB and 3HA. If only the splice form selection drives the switch in quinone type then 4HB and 3HA levels should not change drastically.

We do indeed intend to carry out metabolomics analysis on the different *C. elegans* strains. We propose to add the following text to address this point.

“While these engineered changes could potentially affect RQ levels via some indirect mechanism such as alterations in the levels of 4HB and 3HA, it is unclear what mechanism would drive those metabolic rewirings. We thus suggest that the more parsimonious explanation is the one we propose here: that COQ-2a and COQ-2e have different substrate preference due to the large change to the core of the enzyme and that this is what drives the shift from UQ to RQ synthesis.”

3) Titrate KCN in Figure 5 C experiment and also feed the helminths with 3HA. Data show that even the UQ-specific form can make a small amount of RQ. Is it enough to survive lower KCN treatment and can it be boosted by 3HA feeding?

We have titrated the KCN and show these data now in Figure 5—figure supplement 1, we thank the reviewers for asking for this and should have included this initially. Doses of KCN below ~150 µM fail to drive worms into RQ-dependent metabolism so we focus on doses above 150 µM KCN and we can clearly see that the *coq-*∆*2e* strain is not distinguishable from the *kynu-1* mutant strain that makes no detectable RQ (our previous paper https://elifesciences.org/articles/48165). We therefore conclude that the *coq-*∆2e strain behaves like a RQ-deficient strain in our assays so while it may make a very small amount of RQ, this is insufficient for survival. We have not tried to feed worms with large amounts of 3HA to try to drive the *coq*-∆2e strain to make increased levels of RQ for two reasons. First, it didn’t seem to us to affect the conclusions — that the splicing of *coq-2* affects the balance between UQ and RQ synthesis. Would the finding that adding a massive excess of 3HA could drive some small change in RQ synth in the *coq*-∆2e strain affect that central conclusion? In our eyes it would not. The second is that 3HA connects with other pathways in both the bacterial food and the worms and so we felt it was not a clean experiment — it would be hard to ascribe changes specifically to *coq-2* and changes in bacterial diet and metabolic state can have major outcomes on RQ-dependent metabolism and worm phenotypes. On this line, we tried something along these lines first by supplementing bacteria with additional Trp — this caused the worms fed on these supplemented bacteria to arrest and die. This was due to some metabolic change in the bacterial food since if we did the same with metabolically inert heat-killed bacteria, we do not see any obvious toxicity of Trp. However, worms grow substantially worse on the heat killed bacteria, adding yet another confounder. We therefore hope the reviewers are happy that we have not done this specific experiment and that they are satisfied with the figure supplement.

4) Can the e-exon, which switches COQ2 to make more RQ, be found in any other organism? It was not clear in the text and methods if the sequence was "BLASTed" against all organisms to see if the e-exon exists in any non-Nematodes.

We apologize if this was unclear in the paper — this is absolutely central to the findings. The e-exon is ONLY present in the species that make RQ — helminths, molluscs and annelids. None of the other animals we examined encode such an exon in their genome and we originally stated “This gene structure is only seen in species that make RQ — we find no evidence in any available data for similar alternative splicing in any mammalian hosts (human and mouse are shown as representatives in Figure 3) or in other lineages that lack RQ, such as yeasts.” We agree that this could be made stronger and so now write that “we only find evidence for alternative splicing between a coq-2a form and a coq-2e like form in animals that make RQ. Crucially, no mammalian hosts show any evidence for this kind of alternative splicing of their coq-2 orthologues, nor do they have an e-like exon either in their gene predictions or genome sequences or in any available RNA-seq or direct transcriptome sequence data (human and mouse are shown as representatives in Figure 3). Note that the RNA-seq datasets are much deeper and more extensive in humans and mouse than in helminths, molluscs or annelids, so this lack of evidence for such alternative splicing is unlikely to be due to a failure to detect such events due to low coverage transcriptome data. We conclude that the e-exon and the mutually exclusive alternative splicing of coq-2 is unique to animals that make RQ and is not found in any other species.”

5) Analyze the in silico model and do binding simulations to explain why F204L and A243S changes would so strongly dictate precursor selection. In the human system, COQ2 can use many modified headgroups so why two mutations make it so specific?

This is a fantastic question and we are beginning to undertake binding studies on COQ-2 and several other enzymes. However, we note that the structure we used to model the helminth COQ-2 structures is very evolutionarily distant — there are only 2 available COQ-2 structures and both are from Archaea. This means that while the structure is likely to be substantially correct, the precise spatial location of each side chain is not guaranteed with enough precision to do any reasonable modeling of binding (at least according to expert colleagues!). Thus while the location of the key helices that change from COQ-2a to COQ-2e is broadly correct, the precise positioning of the 2 side chains highlighted in the paper cannot be known with sufficient precision. It’s frustrating, and we would love to know more but that’s the best we can do at this stage and hope the reviewers understand the limitations of the available structures.

Editorial points:The text should also be revised to address the following:– Throughout: The text could be improved to correctly describe the results of the experiments. For example, in the final conclusion authors say that "A switch between two mutually exclusive exons changes the core of the COQ-2 enzyme and switches it from generating UQ precursors to RQ precursors" however, Figure 5C shows that RQ-specific form makes both RQ and UQ, and also UQ-specific form can make a small amount of RQ. Therefore, it is somewhat misleading to suggest that individual splice forms can make only UQ or RQ.

We understand there was a lack of clarity in some of our explanations. We updated the summary paragraph at the end of the Results section to include the sentence:

“A switch between two mutually exclusive exons changes the core of the COQ-2 enzyme and switches it from primarily generating UQ precursors to primarily RQ precursors. We propose that this switch ultimately results in a shift in quinone content from high UQ in aerobic conditions to high RQ in anaerobic conditions.”

Prior to this paragraph we added another conclusion sentence:

“We conclude that increased inclusion of the *coq-2e-*specific exon correlates with increases synthesis of RQ during the parasite life cycles.”

We also rewrote the last paragraph in the Introduction to further clarify these concerns:

*“*In this study, we reveal that two variants of COQ-2, derived from alternative splicing of mutually exclusive exons, are the key for the choice to make RQ or UQ. We find that one of the mutually exclusive exons is only found in the genomes of animals that make RQ and that its inclusion remodels the core of the COQ-2 enzyme. We show that the removal this exon abolishes RQ biosynthesis in *C. elegans*. Finally, we find that inclusion of this RQ-specific exon expression is increased in the stages of the life cycle of parasites where they encounter hypoxic conditions and where they increase RQ production, while the alternative exon is increased in normoxic life stages. We thus conclude that alternative splicing of COQ-2 is the key mechanism that causes the switch from UQ to RQ synthesis in the parasite life-cycle.”

– Introduction paragraph three: Maybe just start sentence with "RQ biosynthesis in animals requires precursors derived from tryptophan" (the evidence in Stairs is not convincing, it could be bacteria in the medium), the situation in microbes does not bear upon these finding. Bernet et al. is convincing, but concerns the R. rubrum gene.

This sentence was simplified. The protist example and the Stairs paper were removed from the sentence. There is a striking contrast between RQ biosynthesis in bacteria and animals and this will be the topic of an upcoming review that two of our co-authors are writing for BBA Bioenergetics. “In contrast to bacteria where RQ derives from UQ (Bernert et al., 2019; Brajcich et al., 2010), RQ biosynthesis in animals requires precursors derived from tryptophan (Figure 1B).”

– Please explain the conversion of the amino group in PABA to the hydroxyl group in HHB in Figure 2C.

The figure was changed to show prenylation of pABA or 4HB gives HHB or HAB, respectively. This was clarified in the figure caption:

“Prenylation is facilitated by Coq2 to form 3-hexaprenyl-4-hydroxybenzoic acid (HHB) or 3-hexaprenyl-4-aminobenzoic acid (HAB), where n = 6. Further functionalization of these intermediates occurs through a CoQ synthome (Coq3-Coq9 and Coq11) to yield UQ.”

– Introduction paragraph four: summarize the findings and restate the Abstract (mollusc and mollusk spelling occur). The meaning of "independent evolution" in is unclear.

Spelling was changed to mollusc throughout.

The statement on “Independent evolution” was removed from the Introduction and is addressed in the final paragraph of the Discussion.

– Figure 1 legend: ETF is not shown in the figure. For malate dismutation to operate (which it does, the endproduct stoichiometry is correct), fumarate reductase (CII) is the RQ oxidant (Müller et al., MMBR 2012).

ETF was removed from the figure caption. We would like to maintain Quinone-coupled dehydrogenases (QDH) in the scheme, since QDHs are broader than just complex II (it also includes quinone-dependent dehydro-orotate dehydrogenase and flavoprotein electron-transfer dehydrogenase). We think that something general is more appropriate. In any case, we included Complex II and fumarate as examples in the caption. For clarity, Panel A was rewritten as follows:

“In aerobic metabolism, ubiquinone (UQ) shuttles electrons in the ETC from Complex I and quinone-coupled dehydrogenases (QDHs), such as Complex II. These electrons are ultimately transferred to oxygen. In anaerobic metabolism, rhodoquinone (RQ) reverses electron flow in QDHs and facilitates an early exit of electrons from the ETC on to anaerobic electron acceptors (Ae^-^A), such as fumarate.”

– Subsection “Regulation of the alternative splicing of coq-2 in helminths” paragraph four: the data show that animal species

This was changed.

– Discussion paragraph three: all animal lineages

This was changed.

– Discussion: The authors favour independent origin of this mechanism and the exons. One needs to spell out what that means. That would mean that sequence homologous exons arose independently in independent lineages, which would mean that e arose from a each time, which is possible, and that e always arose in front of a in each of the 14 lineages surveyed, a 1/16000 proposition if all of the events were independent. Of course, if e arose from a in the common ancestor of these lineages followed by many differential losses in aerobic lineages, then we would have normal Darwinian evolution and no need for LGT or independent origins (googling differential gene loss in eukaryotes or evolution by gene loss returns a lot of hits, the evidence for the widespread occurrence of loss in eukaryotes is uncontroversial). Also possible. Of course, it is also possible that the alternative RQ specific exon here arose only once in one of these animals has been passed around via LGT among eukaryotic lineages (Leger et al., 2018) and specifically inserted into the COQ2 gene of the anaeobes in the conserved position (always in front of the UQ exon) in adaptation to anaerobic niches.– What if we consider the possibility that a single origin and loss, not independent origins (by whatever mechanism) are the cause of the homologous exons (and exon order). What use would RQ be to the common ancestor of molluscs annelisds and nematodes? It would be essential.

We thank the editor for these thoughtful comments. We concur with the editor that we were overenthusiastic in supporting the independent evolution of coq-2 alternative splicing in different lineages. We changed our somewhat biased suggestions to a more balanced view, in which we discuss alternative evolutionary scenarios. We conclude that our current data can neither support an ancestral animal origin of RQ biosynthesis followed by independent losses, or independent animal origin of RQ biosynthesis. We added the following statement to the Discussion:

“It is unclear that our findings here definitively answer this but we do note that there is a precedent for a mutually exclusive splicing event that has arisen independently in helminths, annelids, and molluscs in the gene mrp-1 (Yue et al., 2017).”

We extensively modified the last paragraphs of the Discussion, which now reads:

*“*This example indicates it is at least plausible that alternative splicing of *coq-2* might have originated independently in helminths, annelids, and molluscs, but we do not believe that our current data can distinguish between an ancestral animal origin of RQ biosynthesis followed by independent losses, or independent animal origin of RQ biosynthesis due to an “*mrp-1*-like” independent evolution of alternative splicing of *coq-2*. Future studies should shed on light on this issue. Whatever the evolutionary scenario, our results show that a simple innovation of a mutually-exclusive alternative splicing event can have a profound effect rewiring animal metabolism and it will be interesting to establish whether alternative splicing of COQ-2 is sufficient to drive RQ biosynthesis, or whether additional innovations are needed.”

– Animal aroses and diversified during a phase of Earth history in which oxygen was still low. Geologists have been saying for 20 years that the Proterozoic was anoxic (see Figure 1 in Lyons et al., 2014 or Figure 2 in Catling and Zahnle, 2020). Palaeontologists have been saying that animals arose and diversified in the Ediacaran (that is, Precambrian; <540 MY ago), nematode-mollusc divergence (do Reis et al., 2015) or the nematode annelid divergence (Parfrey et al., 2011). When I find 540 MY ago on the timescales in Lyons et al. and Catling and Zahnle, oxygen is low. Moreover, during the one billion years of eukaryote evolution before that, oxygen was low too. Some of us have been saying for 20 years (Zimorski et al., 2019; Gould et al., 2019) that survival in anaerobic environments was a normal thing that eukaryotes did for over a billlion years of evolution before life on land above the soli line. In that interpretation, the expectation would be that mechanisms of aerobic-anaerobic switching exist that are conserved across annelids, molluscs and nematodes (including free living and parasitic forms). My goodness, a Nobel Prize in Physiology and Medicine went for HIF last year, which senses low oxygen in animals. HIF is conserved across all animals. Did the HIF-dependent oxygen sensing cascade arise independently in all animals? Not likely. Was the HIF pathway laterally transferred? Not likely. I would think that the authors might wish to at least entertain the possibility that the anaerobic physiology that they see in these in animals (and eukaryotes) is conserved from the low oxygen and anaerobic past of animal and eukaryotic lineages. Is it possible that HIF sends the signal that regulates the RQ response? Yes, but I know of no evidence to suggest that, and there are HIF-independent possibilites. They mention Hochachka's work on vertebrates and the goldfish ethanol fermentation example. The vertebrate mechanisms (anaerobiosis in humans will work for about a minute) are not UQ dependent. This is the author's paper and they should write what they want. But if they say independent origins, that interpretation carries a lot of corollaries, none of which are spelled out in the paper, but are spelled out here. Of course, maybe 20 years of geochemical data on late oyxgen, the known age of fossil animals, the conservation of HIF and the conservation of pathways for eukaryote anaerobe survival are all wrong.

The issues raised by the editor are fascinating. However, we have not included them in the discussion, since they remain highly speculative.

– Why does C. elegans make both RQ and UQ? A paragraph in the Discussion would be helpful, even if it is speculative.

We thank the editor for this comment. This is now discussed. We include the following text in the Discussion:

*“C. elegans* is a free-living nematode that does not face extended anaerobic conditions as an obligate part of its life-cycle. However, *Pseudomonas aeruginosa*, a natural pathogen of *C. elegans*, uses cyanide to kill the worm (Gallagher and Manoil, 2001). Cyanide blocks the conventional ETC at complex IV preventing the use of oxygen as a final electron acceptor. We speculate that in *C. elegans,* the alternative ETC may be an adaptive strategy to withstand a transient cyanide stress from pathogens. For parasitic helminths like *Ascaris*, the need to respire anaerobically is critical for their life cycle since they must survive for long periods in the anaerobic environment of the human gut.”

– Do the other enzymes involved in RQ or UQ biosynthesis change in expression under anaerobic conditions?

This is a very interesting question that we would like to explore. However, we do not have data to provide for this manuscript.

[Editors' note: further revisions were suggested prior to acceptance, as described below.]

Revisions:Reviewer #2:The authors have put worth a reasonable revision that provided very little new data, but nonetheless somewhat bolsters their compelling model. Given that the authors haven't, and perhaps can't, provide certain validations of their data, I think the title and some language should be toned down. In the end, both splice forms can make both quinones, but at a different ratio. Also, both splice forms are often co-expressed. The levels of 4HB and 3HA headgroups were not measured so how they impact levels of the quinones is unknown. It will require a significant amount of biochemical data to fully understand the exact differences between the COQ-2a and COQ-2e splice forms. However, it is quite unlikely that the general model put forth is incorrect.Reviewer #3:I continue to think this is a nice study that should be published in eLife. The authors did a nice job in their rebuttal and I am satisfied overall. I agree that RNA-seq is totally fine, given the depth and precision of the method. However, I find parts of the Discussion a bit unsatisfying. For instance, what is the function of alternative splicing of mrp-1? There is no reference for the statement that this transporter transports B12 into the mitochondria. Also, it is not clear that vitamin B12 is required for RQ synthesis in C. elegans, even though it is stated. Finally, the relationship with branched chain fatty acid synthesis is unclear to me. However, clarifying this is up to the authors and shouldn't prevent acceptance of the work.

We have tried to address the additional comments raised by the two reviewers regarding our previous revised manuscript. Specifically, we have changed the title to something that is more precise and is formally correct and have toned down some of the language as requested. For example instead of stating that the COQ-2e isoform is required for RQ synthesis, we now say that it is required for efficient RQ synthesis since the other isoform does indeed make an incredibly low level of RQ. We have also removed a small section in the Discussion to address the question raised by the other reviewer about *mrp-1* and the B12 involvement in RQ-dependent metabolism. It was not central for the paper or conclusions in any way and we actually prefer it as it reads now, so we thank this reviewer for that suggestion.

**References**

dos Reis et al. (2015) Uncertainty in the timing of origin of animals and the limits of precision in molecular timescales. Curr Biol 25, 2939-2950,

Parfrey LW, Lahr DJ, Knoll AH, Katz LA. 2011. Estimating the timing of early eukaryotic diversification with multigene molecular clocks. Proc Natl Acad Sci USA 108: 13624-13629.

Catling and Zahnle, Sci. Adv. 2020;6: eaax1420

Lyons TW et al. 2014. The rise of oxygen in Earth's early ocean and atmosphere. Nature 506: 307-315.

Leger et al. 2018. Demystifying eukaryote lateral gene transfer. BioEssays, 40, 1700242

Gould SB et al. Adaptation to life on land and high oxygen via transition from ferredoxin- to NADH-dependent redox balance. Proc. Roy. Soc. Lond. B. 286: 20191491 (2019)

Zimorski V et al. Energy metabolism in anaerobic eukaryotes and Earth's late oxygenation. Free Radicals Biol. Med. 140:279-294 (2019).